# *SUCLA2* mutations cause global protein succinylation contributing to the pathomechanism of a hereditary mitochondrial disease

Philipp Gut [1,2,17✉], Sanna Matilainen[3,17], Jesse G. Meyer[4,14,17], Pieti Pällijeff[3], Joy Richard[2], Christopher J. Carroll[5], Liliya Euro [3], Christopher B. Jackson [3], Pirjo Isohanni[3,6], Berge A. Minassian[7,8], Reem A. Alkhater[9], Elsebet Østergaard[10], Gabriele Civiletto[2], Alice Parisi[2], Jonathan Thevenet[2], Matthew J. Rardin[4,15], Wenjuan He[1], Yuya Nishida[1], John C. Newman [1], Xiaojing Liu[11,16], Stefan Christen[2], Sofia Moco [2], Jason W. Locasale [11], Birgit Schilling [4,18✉], Anu Suomalainen [3,12,13,18✉] & Eric Verdin [1,4,18✉]

Mitochondrial acyl-coenzyme A species are emerging as important sources of protein modification and damage. Succinyl-CoA ligase (SCL) deficiency causes a mitochondrial encephalomyopathy of unknown pathomechanism. Here, we show that succinyl-CoA accumulates in cells derived from patients with recessive mutations in the tricarboxylic acid cycle (TCA) gene succinyl-CoA ligase subunit-β (*SUCLA2*), causing global protein hyper-succinylation. Using mass spectrometry, we quantify nearly 1,000 protein succinylation sites on 366 proteins from patient-derived fibroblasts and myotubes. Interestingly, hyper-succinylated proteins are distributed across cellular compartments, and many are known targets of the ($NAD^+$)-dependent desuccinylase SIRT5. To test the contribution of hyper-succinylation to disease progression, we develop a zebrafish model of the SCL deficiency and find that SIRT5 gain-of-function reduces global protein succinylation and improves survival. Thus, increased succinyl-CoA levels contribute to the pathology of SCL deficiency through post-translational modifications.

[1] Gladstone Institutes and University of California, San Francisco, CA 94158, USA. [2] Nestlé Research, Institute of Health Sciences, EPFL Innovation Park, 1015 Lausanne, Switzerland. [3] Research Program of Stem Cells and Metabolism, Biomedicum Helsinki, University of Helsinki, 00290 Helsinki, Finland. [4] Buck Institute for Research on Aging, Novato, CA 94945, USA. [5] Genetics Research Centre, Molecular and Clinical Sciences, St. George's, University of London, London, UK. [6] Department of Child Neurology, Children's Hospital, University of Helsinki and Helsinki University Hospital, Helsinki, Finland. [7] Program in Genetics and Genome Biology, The Hospital for Sick Children, Institute of Medical Science University of Toronto, Toronto, Ontario, Canada. [8] Division of Neurology, Department of Pediatrics, University of Texas Southwestern, Dallas, TX, USA. [9] Johns Hopkins Aramco Healthcare, Dhahran, Saudi Arabia. [10] Department of Clinical Genetics, Copenhagen University Hospital Rigshospitalet, 2100 Copenhagen, Denmark. [11] Department of Pharmacology and Cancer Biology, Duke University School of Medicine, Durham, NC, USA. [12] Neuroscience Center, HiLife, University of Helsinki, 00290 Helsinki, Finland. [13] HUSlab, Helsinki University Hospital, 00290 Helsinki, Finland. [14]Present address: Department of Biochemistry, Medical College of Wisconsin, Milwaukee, WI 53213, USA. [15]Present address: Amgen, Thousand Oaks, CA 91320, USA. [16]Present address: North Carolina State University, Raleigh, NC 27607, USA. [17]These authors contributed equally: Philipp Gut, Sanna Matilainen, Jesse G. Meyer. [18]These authors jointly supervised this work: Birgit Schilling, Anu Suomalainen, Eric Verdin. ✉email: philipp.gut@rd.nestle.com; bschilling@buckinstitute.org; anu.wartiovaara@helsinki.fi; everdin@buckinstitute.org

Lysine side-chains of proteins can be modified by small acyl groups, and these posttranslational modifications (PTMs) can affect protein function through various mechanisms, including changes in structural conformation, protein activity, protein-protein interaction, or subcellular localization[1]. Protein acylation can be enzymatic; histone acetyl-transferases (HATs) regulate gene expression by transferring the acetyl group from acetyl-CoA to defined lysine residues on histones, thereby opening up chromatin and allowing access to the transcriptional machinery[2]. However, we first proposed that acyl-CoA metabolites are highly reactive chemical species that likely modify lysine side chains in the absence of a catalyst, particularly at the basic pH of the mitochondrial matrix[3]. This prediction was validated experimentally in vitro[4–9]. In fact, although protein acylation is abundant in the mitochondria, no acyl-transferase has been found within mitochondria, suggesting that non-enzymatic acylation is the primary source of mitochondrial acyl-lysine PTMs. Several acyl-groups, such as acetyl-CoA, succinyl-CoA, propionyl-CoA, glutaryl-CoA and crotonyl-CoA, are produced in mitochondria by catabolic metabolism[4,10–12], although a large number of acylated proteins are located in cellular compartments other than the mitochondria[6,9]. Pathways involved in the generation or utilization of mitochondrial acyl-CoAs include, among others, the tricarboxylic acid (TCA) cycle, fatty acid oxidation, urea detoxification, ketogenesis, and initial steps of cholesterol synthesis. Importantly, these pathways are extensively modified by acylation and are the main targets of the mitochondrial sirtuins SIRT3, SIRT4 and SIRT5 that function as lysine deacy-lases (KDAC)[12,13]. The reciprocal interplay of reversible, non-enzymatic acylation and removal of a subset of these PTMs by sirtuins is believed to have co-evolved as an important regulatory mechanism of intermediary metabolism[6,12,13]. Moreover, non-physiological increases of protein acylation disrupt metabolic pathways and are termed carbon stress[6,7,11,12,14,15].

The sirtuin family member SIRT5 is located in the cytosol and mitochondria where it removes negatively charged acylcarboxyl moieties, including succinyl-, malonyl-, and glutaryllysine modifications[16–19]. The extent to which loss of SIRT5 function leads to increased levels of protein acylation is likely determined by substrate availability in the respective compartment. For example, while loss of SIRT5 in mice affects both mitochondrial and cytosolic protein malonylation, most modified proteins reside in the cytosol where malonyl-CoA is an abundant intermediate of triglyceride synthesis. Consequently, nearly all enzymes of the glycolytic pathway carry malonyllysine modifications that lead to a reduced glycolytic flux in response to loss of SIRT5 function[18].

Unlike cytosolic malonylation, protein succinylation is more prevalent in the mitochondria, where succinyl-CoA is a metabolite of the TCA cycle that yields succinate via the enzyme succinyl-CoA ligase (SCL)[9,17]. Mitochondrial protein succinylation regulates the urea cycle, ketogenesis, fatty acid oxidation and TCA cycle flux by modulating the key enzymes of these processes[16,17]. At an organismal level, whole-body knockout of Sirt5 in mice causes cardiomyopathy in response to chronic pressure overload and during aging, suggesting that increased succinylation causes pathologies[20,21]. Increased levels of succinylation have been well studied in conditions of SIRT5 loss-of-function, but much less is known about how increased levels of substrates affect global protein succinylation. Research with yeast indicates that disrupting the succinyl-CoA ligase reaction increases global protein succinylation[9]. However, yeast lack enzymes with desuccinylase activity, and it is not clear if increased succinyl-CoA levels would overcome SIRT5-mediated desuccinylation and have functional consequences.

SCL deficiency is a mitochondrial DNA (mtDNA) depletion syndrome with methylmalonic aciduria (OMIM#612073 and #245400) that manifests with a Leigh/Leigh-like encephalomyopathy[22–24]. Altogether, more than 20 patients with mutations in the gene encoding the α-subunit, SUCLG1, and 50 patients with mutations of the gene encoding the ATP-specific β-subunit, SUCLA2, have been reported. Disease-causing mutations in the GTP-specific β-subunit, SUCLG2, have not been described. Patients with SUCLA2 mutations typically manifest in early childhood with a progressive encephalomyopathy, including hypotonia, dystonia, and sensorineural deafness. The molecular characteristics include methylmalonic aciduria and mtDNA depletion in muscle tissue, but these are not present in all patients. Disease progression varies from an early-onset rapidly progressive phenotype, leading to early death before adulthood, to a slowly progressive disease[22,23]. The rapidly progressing form of the disease is associated with mutations that cause instability or absence of functional SUCLA2 protein or completely inactive ADP phosphorylation, whereas mutations associated with the less fulminant disease course commonly lead to partially impaired enzymatic function[24]. Mutations in SUCLG1 cause a similar, often rapidly progressive phenotype. Some patients also manifest with hepatopathy and hypertrophic cardiomyopathy[22,25]. To date, studies of the molecular pathomechanisms of SCL deficiency have focused on the primary defect in the TCA cycle. The mechanistic basis of the mtDNA depletion, the typically abnormal carnitine profile and the disturbed leucine metabolism in some patients, however, remain unknown.

In this study, we test the hypothesis that impairment of SCL activity in patients with mitochondrial disease due to mutations in SUCLA2 leads to increased levels of succinyl-CoA as a donor for succinylation reactions. Indeed, we find that patient-derived cells show increased succinyl-CoA levels that are associated with a global increase of site-level protein succinylation. Using mass spectrometry, we generate a detailed list of these pathological succinylation targets. These proteins are involved in many important pathways for metabolic control in mitochondria, but also in the nucleo-cytosolic compartment. Many of these pathologically succinylated proteins are regulated by SIRT5. We therefore test whether the Sirt5 gain-of-function modifies global protein succinylation and affects the pathology in a novel zebrafish model of SCL disease. We find that increased Sirt5 activity reverses the build-up of protein succinylation and partially restores oxidative metabolism. In addition, survival is improved by sirt5 overexpression, an effect that is independent of effects on mtDNA levels. In sum, we describe global protein succinylation as a biochemical hallmark in SCL patients, and we propose that disruption of affected pathways by uncontrolled PTMs contributes to the complex clinical symptoms.

## Results

**Succinyl-CoA ligase deficiency causes protein succinylation.** We and others reported that TCA cycle enzymes carry succinylation modifications in livers from $Sirt5^{-/-}$ mice[17,26]. We applied a comparative analysis approach to these data sets. Among all reported succinylated proteins, TCA cycle enzymes are enriched for succinylation modifications when all cellular proteins are rank-sorted by the number of acylated sites corrected by the abundance of those proteins (Wilcoxon test $p$-value = 6E-4, Supplementary Fig. 1a). Most TCA cycle subunits carry several succinylation sites (Supplementary Fig. 1b). This finding is consistent with the concepts that the abundance of succinyl-CoA in the mitochondria drives non-enzymatic acylation and that the TCA cycle is the predominant source of succinyl-CoA[4,9]. In the absence of the desuccinylase SIRT5, the abundance of many succinylation sites on TCA cycle proteins increased (Fig. 1a).

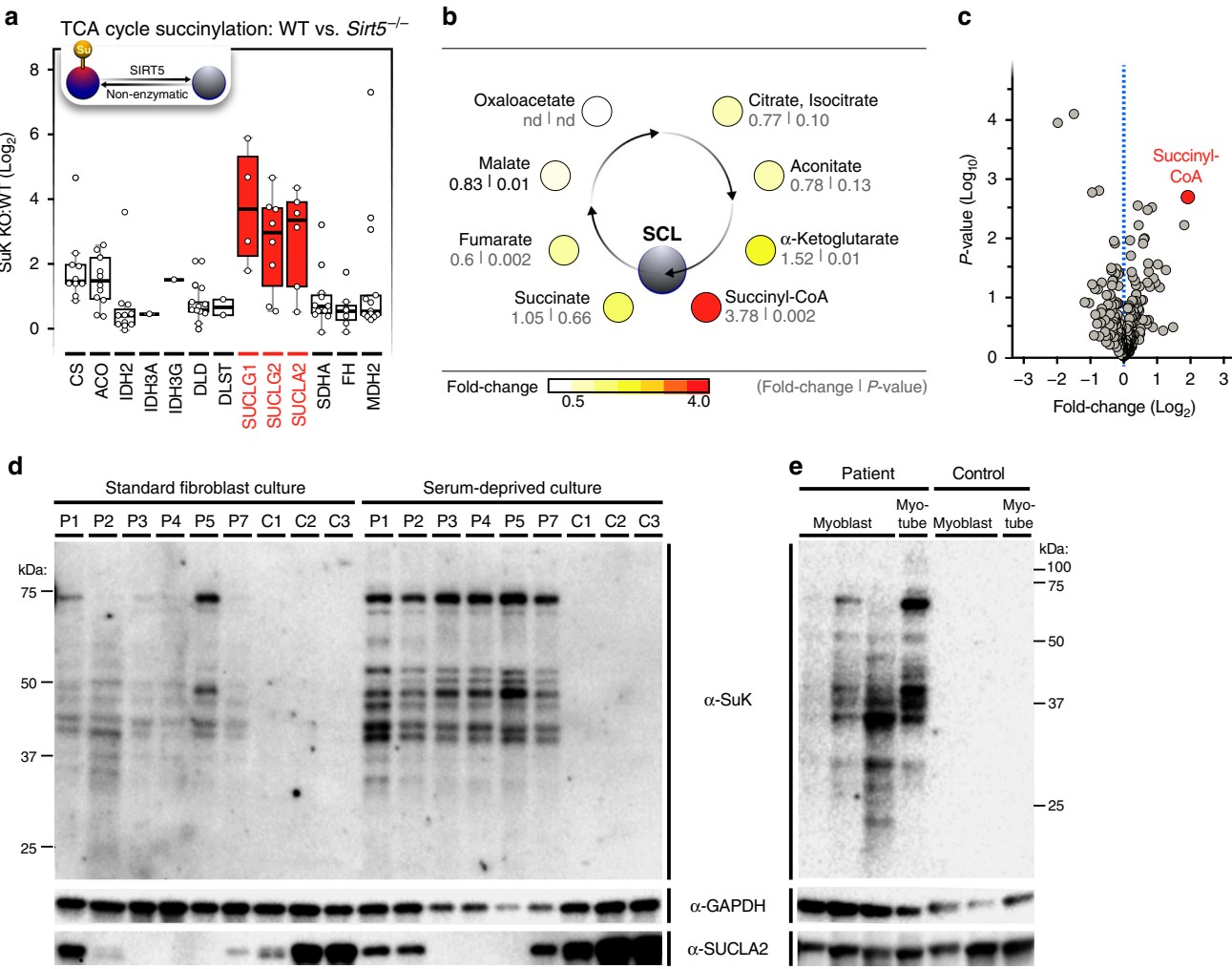

**Fig. 1 SCL deficiency causes global protein hyper-succinylation. a** Quantitation of murine liver proteome succinylation shows TCA cycle enzymes are targets of lysine succinylation in mice (C57/BL6, $n = 10$ wild-type, $n = 10$ $Sirt5^{-/-}$). The SCL complex shows higher fold-changes than other TCA cycle enzymes. "Su" in the scheme indicates a succinylation modification that is removed by SIRT5. The boxplot shows the median, the first to third quartile, the 1.5x interquartile ranges, and outliers. **b** Untargeted metabolomics show more succinyl-CoA and α-ketoglutarate in patient-derived fibroblasts than in control fibroblasts. The color scale indicates fold-change. **c** Volcano plot of metabolite changes in fibroblasts from SCL patients and control fibroblasts. The red dot shows succinyl-CoA. **d** SDS-PAGE and western blot analysis of lysine succinylation in fibroblasts cultured in standard culture conditions (proliferative condition) and cells cultured for 5 days in low-serum conditions (non-proliferative condition). Patients (P1–P7) carry disease-causing mutations in *SUCLA2*. Controls (C1–C3) are fibroblasts from age-matched patients with other mitochondrial diseases. **e** SDS-PAGE and western blot analysis of lysine succinylation during myoblast differentiation in myotubes (arrow indicates stages of differentiation) of patient with SCL deficiency due to Asp333Gly mutation in *SUCLA2*. Control is an age-matched patient with a non-mitochondrial disease. GAPDH was used as a loading control, and residual SUCLA2 levels in patient samples are shown. Western blot images are representative results from 10 different independent experiments with similar results. The *p*-values were calculated with Welch two-sample t-test assuming unequal variance. Source data are provided as a Source Data file. WT = wild-type; SuK = succinyl-lysine, nd = not determined.

These data suggest that TCA cycle proteins involved in succinyl-CoA production or utilization are uniquely susceptible to reversible lysine succinylation.

From data published by Rardin et al.[17], among all SIRT5-regulated succinylation sites on the proteins in the TCA cycle, succinylation sites on SCL subunits (SUCLG1, SUCLA2 and SUCLG2) had the largest fold-changes (Fig. 1a and Supplementary Fig. 1c, d). Distributions of succinylation sites on TCA cycle proteins were different from each other (log2(fold change), ANOVA *p*-value = 1E-7); succinylation site changes on all three SCL subunits (SUCLA2, SUCLG1, SUCLG2) were statistically significantly higher than sites on the other TCA proteins FH, IHD2, DLD, and SDHA (Tukey's post hoc test adjusted *p*-value <0.02). SUCLA2 and SUCLG2, the catalytic subunits of SCL, each carry six succinylated lysines within their nucleotide

grasp-domains, and these residues showed SIRT5-dependent changes up to 25-fold (Supplementary Fig. 1c, d). We mapped the detected succinylation sites to the protein structure of SCL and found Lys66 in SUCLG1 subunit and Lys108, Lys116 and Lys143 in the SUCLA2 subunit of SCL. These lysines are highly conserved among species (Supplementary Fig. 1e). No variants of Lys108, Lys116 and Lys143 of SUCLA2 were found in the Exac database (The Exac database has been moved to gnomad: https://gnomad.broadinstitute.org/, accessed 20.8.2018). One heterozygotic allele of a synonymous nucleotide change affecting Lys66 of SUCLG1 was found among 119,390 alleles. Lys66 locates at 3.7 Å from bound succinyl-CoA in SUCLG1. This lysine is critical in substrate binding as it is positioned close to the negatively charged phosphate groups of succinyl-CoA. Succinylation of this lysine is expected to change the local charge, resulting in

**Table 1 Characteristics of SCL patients and respective gene mutations.**

| Patient | P1 | P2 | P3 | P4 | P5 | P6 | P7 |
|---|---|---|---|---|---|---|---|
| Gender | M | M | F | M | F | M | F |
| Age at disease onset | 5 mo | 6 mo | 2 mo | 5 mo | 5 mo | 2 mo | 12 mo |
| Proteomics analysis | + | | | + | | | + |
| Metabolomics analysis | + | + | + | + | + | + | + |
| Nucleotide change | c.998A > G | c.998A > G/13q14 deletion (1.54 Mb) | c. 534 + 1G > A | | | | c.1219C > T |
| Amino acid change | Asp333Gly | Asp333Gly/13q14 deletion | Splice site mutation, skipping of exon 4 | | | | Arg407Trp |
| OXPHOS defect | Normal (muscle, histochemistry and biochemical analysis) | Partial deficiency of CI + CIII (muscle, biochemical analysis) | Slight decrease of CI, CIII and CIV (muscle, SHS immunoblot); normal/ slight decrease of CIV (fibroblasts, SHS immunoblot) | | | | Normal (muscle, histology, EM and OXPHOS analysis) |
| References | | 24 | 35,59 | | | | This report |

Mutations and clinical data of patients 1–6 were reported in the indicated references. Patients included in the proteomics and metabolomics analyses indicated with +. M = male, F = female, mo = months. P = patient.

decreased affinity for succinyl-CoA (Supplementary Fig. 1f, left panel). Lys108, Lys116 and Lys143 in SUCLA2 are located on the edge of the ADP-binding cleft. These lysine residues are important for substrate binding by interacting with negatively charged Glu193, Asp194 and Glu198 on the opposite side of the ADP-binding cleft. Succinylation of these conserved lysines changes the local positive charge to negative, likely resulting decreased affinity for ADP (Supplementary Fig. 1f, right panel). These data suggest that succinylation sites on SCL are important targets of regulation by SIRT5 and that succinylation sites on SCL may function to inhibit SCL as part of a feedback loop.

We next asked whether loss of enzymatic flux through SCL in patients with *SUCLA2* mutations caused an increase in succinyl-CoA levels. Untargeted metabolomics was used to compare fibroblasts from patient and control fibroblasts. The characteristics of each patient-derived cell line are described in Table 1. Metabolomic data showed increased abundance for TCA cycle metabolites upstream of SCL, specifically succinyl-CoA and α-ketoglutarate (Fig. 1b). Notably, succinyl-CoA was the most increased metabolite in fibroblasts from patients with SCL deficiency (Fig. 1c and Supplementary Data 1).

The hallmark of the disease, mtDNA depletion, manifests in patient-derived fibroblasts in non-proliferating, serum-deprived conditions[27], and therefore, we studied the patient-derived cell lines under different culture conditions. Western-blot analysis of non-proliferative serum-deprived fibroblasts and differentiated myotubes from patients with disease-causing mutations in *SUCLA2* showed more global protein succinylation than cells from age-matched controls (~8-fold) and patient myoblasts (~5-fold). Proliferating fibroblasts cultured under standard conditions showed a 1.9-fold increase of global protein succinylation in *SUCLA2*-patients (Fig. 1d, e and Supplementary Figs. 2 and 3).

Thus, high succinyl-CoA levels resulting from reduced SCL enzymatic activity are sufficient to cause accumulation of protein succinylation in post-mitotic cells of patients with SCL deficiency. These data identify global protein succinylation as a biochemical hallmark in *SUCLA2* patients.

**Proteomic analysis of lysine succinylation targets**. To identify and quantify specific lysine residues in proteins that are succinylated when SCL flux is impaired, we used antibody-based enrichment of succinylated peptides and mass spectrometry (MS) as described[28,29] (Fig. 2a). To account for potential changes in protein abundance, we also separately quantified proteins and found very few protein-level changes resulting from SCL deficiency (Supplementary Data 2 for fibroblasts and Supplementary Data 3 for myotubes). Protein quantities were used to correct site-level changes enabling comparison of a pseudo site occupancy.

In fibroblasts, we identified 933 succinylated lysine (SuK) residues distributed across 366 proteins (Fig. 2b, c, Supplementary Data 4). In myotubes, 194 lysine residues in 91 proteins were succinylated; most (86%) were also detected in fibroblasts (Fig. 2b, c, Supplementary Data 5). The majority of SuK sites were found more often in patient samples than controls (fold change >2, FDR < 0.05) (Fig. 2d, e). Most proteins had a single SuK site, but several had multiple modifications (Fig. 2f). We performed gene ontology (GO) term enrichment analyses with the list of succinylated proteins found in fibroblasts to identify the subset of proteins that harbor increases in lysine succinylation due to SCL deficiency. This analysis does not reveal the true population of modified proteins because of analytical bias toward detecting sites on the most abundant proteins[30], but it does reveal the types of proteins we followed in our study. As expected, the TCA cycle Gene Ontology (GO) term was significantly enriched at the top of the list (Fig. 2g). Our data also contained overrepresentation of succinylation sites on proteins involved in mitochondrial energy production, such as ATP biosynthesis, pyruvate metabolism and β-oxidation. Additionally, GO terms representing proteins outside the mitochondria were enriched, such as cell-cell adhesion, protein folding, and glycolysis/gluconeogenesis. A plot of the top 20 proteins sorted by number of lysine marks (Fig. 2h) and fold-change of individual lysine residues (Fig. 2i) showed that we detect many succinylation sites on proteins that reside in different subcellular compartments, including mitochondria, the cytosol and the endoplasmic reticulum. Notably, this subset of sites almost certainly is biased toward the more abundant proteins in the sample and, therefore, does not reflect the true population of sites influenced by SCL deficiency. We expected enrichment of mitochondrial pathways due to their proximity to succinyl-CoA production, since we observed that nearly all succinylation sites increase and we expect this process to be nonenzymatic. Thus, there should be no true enrichment in functional categories. Still,

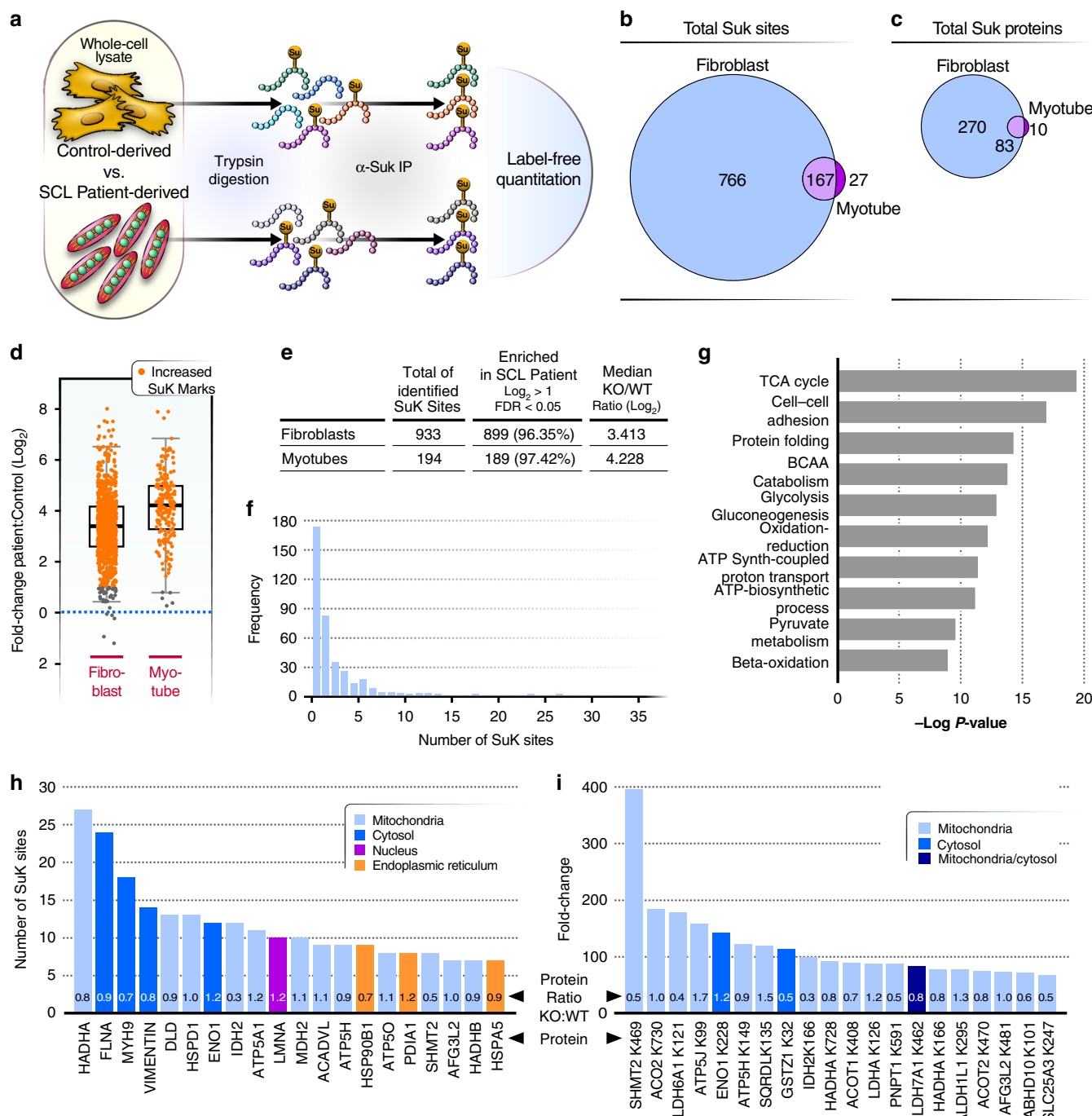

**Fig. 2 Proteomics analysis of protein lysine succinylation. a** Workflow for label-free proteomics quantitation of succinyl-lysine residues in whole-cell lysates of fibroblasts and myotubes from SCL patients and controls. Samples were trypsin-digested. Succinylated lysines were enriched by immune affinity and analyzed by mass spectrometry ($n = 2$ control, $n = 3$ SCL patient-derived fibroblasts; $n = 2$ control, $n = 1$ patient-derived myotubes; A technical replicate of patient-derived myotube samples was used). **b** Venn diagram of total number of succinylated lysine sites identified in fibroblasts and myotubes. **c** Venn diagram of total number of proteins carrying succinyl-lysine sites. **d** Boxplot of fold-changes of succinyl-lysine mean intensities in fibroblasts and myotubes from patients compared to controls. Orange points indicate lysine marks enriched significantly more in patient samples, compared to controls. The boxplot shows the median, the first to third quartile, the 1.5x interquartile ranges, and outliers. **e** Table of identified SuK sites in fibroblasts and myotubes showing total numbers of SuK sites, numbers of SuK sites significantly enriched in SCL patient samples, and the median values of $\log_2$-transformed ratios of patient-derived samples, compared to that from control cells. **f** Distribution of frequencies of succinylated lysine PTMs per detected protein. More than half of the proteins contain only one succinylated lysine. Oher proteins carry more than one SuK mark with a maximum of 27 SuK PTMs. **g** GO biological process term enrichment analysis of hyper-succinylated proteins from the fibroblast samples. Statistical significance was determined using fisher exact test. **h** Barplot of the 20 proteins with the highest numbers of SuK marks identified in our analysis. Highly succinylated proteins are found in different cellular compartments, including mitochondria, the nucleus, the endoplasmic reticulum and the cytosol. Protein ratio indicates levels of proteins in SCL patient-derived cells vs control cells. **i** Barplot of the 20 most dynamically affected succinylation marks in the absence of *SUCLA2* as detected in this analysis. Source data are provided as a Source Data file. SuK= succinyl-lysine; TCA = tricarboxylic acid; BCAA = branched chain amino acid.

these analyses are presented to convey the analytical coverage of our study. In summary, SCL deficiency leads to high levels of reactive succinyl-CoA, which affects proteins of diverse cellular pathways by non-enzymatic lysine succinylation.

**Succinylation targets overlap in SCL and *Sirt5* deficiencies**. The protein with the most succinylation sites, hydroxyacyl-CoA dehydrogenase trifunctional multi-enzyme subunit-α (HADHA), is an integral part of the enzyme complex that regulates β-oxidation and a major target of SIRT3 and SIRT5 (refs. [17,21,31]). Similarly, other proteins with high numbers of succinylation sites (e.g., filamin A (FLNA), myosin heavy chain 9 (MYH9), heat shock protein family D 1 (HSPD1) and AFG3-like protein 2 precursor (AFG3L2)) were found in proteomics analyses of SIRT5 deficiency[26]. However, these latter studies used tissues from different species and with different proteomics workflows. To assess a potential overlap between these two drivers of protein succinylation, hyper-succinylated proteins and sites detected in fibroblasts from patients with SCL deficiency were compared with those in SIRT5 knockout (KO) mouse embryonic fibroblasts by Park et al.[26]; 102 proteins carry at least one succinylated lysine in both conditions, representing 41% of all proteins detected with hypersuccinylation due to SCL deficiency, or 29% of hyper-succinylated proteins detected due to deletion of SIRT5 (Fig. 3a). Since the two studies were carried out in different organisms and mouse protein sequences carry amino acid insertions and deletions relative to human proteins, pairwise sequence alignment of these 102 common proteins was performed to allow site-level comparison. This analysis revealed that 238 sites were in common between the two studies (Fig. 3b, Supplementary Fig. 4, and Supplementary Data 6). Interestingly, the extent of hyper-succinylation due to SCL deficiency was greater than what was observed from mouse embryonic fibroblasts lacking SIRT5 (Fig. 3c).

The sites and proteins we detected are biased toward the most abundant succinylated sites and proteins. However, another useful way to examine the overlap between studies of SCL deficiency and SIRT5 KO is to compare the number of modifications detected per protein. Among these common proteins, the numbers of succinylation sites per protein were correlated between both conditions ($r^2 = 0.54$, $p$-value= 2.2e-16), including filamin A, MYH9, HADHA and vimentin, which were some of the proteins with the highest numbers of succinylation sites in both conditions (Fig. 3d). Because nearly all (98%) of sites increased in levels of succinylation due to SCL deficiency, as would be expected in a non-enzymatic process, the sites we detected in our study likely overlap with those in other data sets related to hyper-succinylation. Therefore, examination of the exact fold-changes resulting from each process on individual lysine sites in both data sets are of interest. To this end, we aligned the amino acid sequences of all identified proteins, taking into account amino acid insertions and deletions that occurred during evolution, and determined those lysines that are modified in both species. Although the comparison of the two different species and methodologies only allows an approximation of fold-changes, we estimate that the median fold-changes across the differential succinylation of most of the top 11 proteins were higher in SCL deficiency than in SIRT5 deficiency (Fig. 3e). Comparing the exact sites of these same 11 proteins under the two conditions, we estimated that almost all sites had lower fold-changes in SIRT5 deficiency than SCL deficiency (Supplementary Fig. 4). Notably, some proteins had sites with a similar regulation in the two studies (e.g., Citrate Synthase and HADHA) (Supplementary Fig. 4, Supplementary Data 6). These data show an extensive analytical overlap between proteins detected to accumulate SuK marks in response to SUCLA2 and SIRT5 deficiencies, which cause increased protein succinylation

by increasing succinyl-CoA levels or decreasing lysine de-succinylation activity, respectively.

**Sirt5 modifies succinylation levels in *sucla2^−/−* zebrafish**. The substantial overlap of pathological succinylation with SIRT5-regulated proteins raises a question: Is pathological succinylation modulated by SIRT5 function? Importantly, increased SIRT5 activity could be applied to reverse global protein hyper-succinylation and could possibly restore the function of those proteins negatively affected by pathological succinylation. To test this hypothesis in a zebrafish model of SCL deficiency, we performed targeted disruptions of *sucla2*, the zebrafish orthologue of *SUCLA2*, and *sirt5*, the zebrafish orthologue of *SIRT5*, with CRISPR/Cas9 genome editing. The vertebrate model zebrafish is amenable to genetic manipulations and the study of energy metabolism, and, important to this study, expresses all seven sirtuin genes[32,33]. In zebrafish, like in mammals, SCL is a heterodimeric complex of three subunits: the non-catalytic subunit suclg1 that dimerizes with suclg2, mainly in the liver and kidney, or with sucla2 in most other organs, including the brain and the skeletal muscle[34]. We selected two lines carrying out-of-frame mutations for *sucla2* and *sirt5* in the germline (Fig. 4a and Supplementary Fig. 5). In addition, we generated a gain-of-function model overexpressing the cDNA of *sirt5* ubiquitously across tissues, *Tg(ubi:sirt5;cryaa:zsgreen1)* (hereafter termed *Tg (ubi:sirt5)*) (Fig. 4a). qPCR analyses of *sucla2* and *sirt5* gene expression confirmed that mRNA was depleted as a result of out-of-frame mutations, leading to nonsense-mediated decay (Fig. 4b, c). qPCR analysis of the *sirt5* transgene expression showed an approximately 7-fold overexpression relative to the level of endogenous *sirt5* mRNA (Fig. 4d).

Using these models, we found that baseline protein succinylation was not detected in *sirt5* mutant zebrafish in the larval stages, which are characterized by proliferation and cell growth. This is similar to what was observed in *Sirt5* loss-of-function cell lines in proliferative conditions[17] (Fig. 4e). By contrast, loss of *sucla2* resulted in an increased succinylation pattern at 7 days post fertilization (dpf) (Fig. 4e). Combined deficiency of *sirt5* and *sucla*2 further increased global protein succinylation significantly, suggesting that succinylation is determined in vivo by both levels of free succinyl-CoA and Sirt5 de-succinylase activity (Fig. 4e). Overexpression of the *sirt5* transgene using *Tg(ubi:sirt5)* zebrafish in the backgrounds of wild-type or *sucla2^−/−* genotypes only partially reversed protein succinylation, suggesting that not all sites affected by pathological succinylation can be efficiently de-succinylated by Sirt5 (Fig. 4f).

In contrast to *sucla2^−/−* larvae, heterozygous carriers of *sucla2* mutant alleles survive into adulthood. To further investigate the occurrence of global protein succinylation in SCL deficiency, we quantified succinylation levels in skeletal muscle of *sucla2^+/−* animals. Surprisingly, even heterozygosity of *sucla2* mutations led to accumulation of protein succinylation in adult tissues that was amplified by *sirt5* deficiency (Supplementary Fig. 6a). In a second set of experiments, we generated transgenic zebrafish that specifically overexpress *sirt5* in skeletal muscle and crossed this line into the *sucla2^+/−* background. Overexpression of *sirt5* led to a significant reduction of protein succinylation levels (Supplementary Fig. 6b). In summary, our data show that *sirt5* is a modifier gene of protein succinylation in SCL deficiency.

**Sirt5 gain of function improves survival**. Genetic disruption of *sirt5* impairs several metabolic pathways, including fatty acid oxidation[17,21,26]. Our analysis also indicates that proteins related to oxidative metabolism and TCA cycle function are affected by succinylation in SCL deficiency. We, therefore, determined

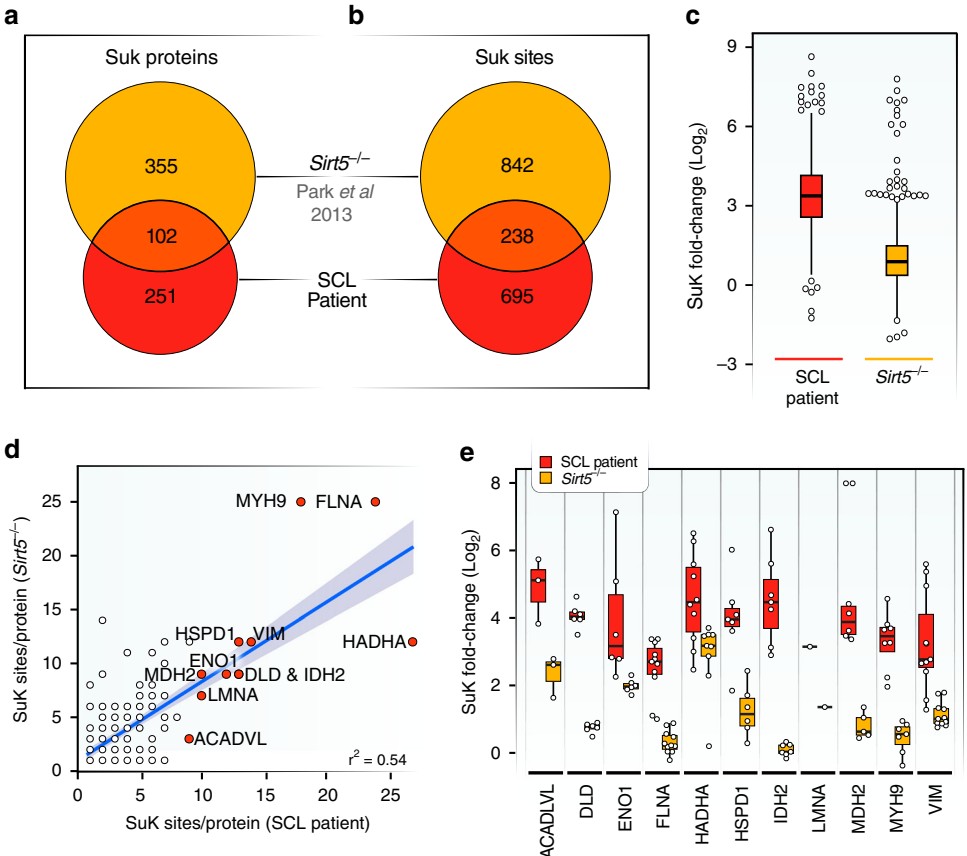

**Fig. 3 Common succinylated targets in SCL and SIRT5 deficiencies. a** Venn diagram of succinylated proteins identified in SCL patient fibroblasts and in mouse embryonic fibroblasts from $Sirt5^{-/-}$ mice. Results from this study were compared to data deposited by Park et al.[26]. **b** Venn diagram showing number of sites that are susceptible to hyper-succinylation in both conditions. Pairwise sequence alignments identified each lysine showing significant changes in succinylation levels in both data sets. **c** Boxplot of fold-changes of lysine succinylation found in SCL-deficient patient cells and mouse embryonic fibroblasts from $Sirt5^{-/-}$ mice ($n = 2$ control fibroblast samples, $n = 3$ SCL patient-derived fibroblast samples; $n = 2$ control fibroblast samples, $n = 1$ patient-derived myotube samples; A technical replicate of patient-derived myotube samples was used). **d** Correlation of the number of suK marks on individual proteins found in data sets from SCL patient and $Sirt5^{-/-}$ fibroblasts. Orange dots indicate the 11 most succinylated proteins in number of modified lysine sites identified in fibroblasts from SCL deficient patients (Total of 189 proteins, $r^2 = 0.54$). Gray band indicates 95% confidence interval. Red dots indicate the 11 proteins with the highest number of succinylation sites in SCL patient fibroblasts. **e** Boxplot showing fold-changes in succinylation of individual lysine residues among the 11 proteins with the highest numbers of suK marks. Note that only sites matched by sequence alignment of our human data to published mouse data are included in this analysis. The boxplots show the median, the first to third quartile, the 1.5x interquartile ranges, and outliers. SuK = succinyl-lysine.

oxygen consumption rate (OCR) of whole zebrafish larvae, using a seahorse analyzer to measure changes in oxidative metabolism. As expected, *sucla2* deficiency led to a strong reduction of basal and maximal, uncoupled respiration (Fig. 5a–c). Whereas over-expression of *sirt5* in wild-type animals did not lead to a significant change in OCR, Sirt5 significantly increased the OCR in *sucla2* mutant zebrafish ($p = 0.0067$) and showed a trend towards increased basal respiration ($p = 0.0725$) (Fig. 5a–c). Next, we determined if this improvement in oxidative metabolism was associated with changes in mtDNA content. We found that, indeed, *sucla2* deficiency led to a significant mtDNA depletion, recapitulating this hallmark of SCL disease (Fig. 5d). Exogenous overexpression of *sirt5* did not affect mtDNA content, compared to the mutant background alone, indicating that changes in oxygen consumption rate are not linked to correction of mtDNA levels (Fig. 5d). Improvement in respiration was also independent of a potential normalization of metabolite levels directly upstream of SCL, including succinyl-CoA and α-ketoglutarate, but instead moderately increased the levels of α-ketoglutarate (Supplementary Fig. 7a–d). These data suggest that increasing Sirt5 activity cannot directly overcome reduced substrate flux through SCL

but has a profound impact on mitochondrial respiratory capacity in SCL deficiency.

Finally, we aimed to determine if *sirt5* affects the health of *sucla2*-deficient zebrafish. In agreement with the strongly reduced capability for oxidative metabolism, *sucla2*$^{-/-}$ larvae die prematurely starting at 10 dpf and do not reach adulthood (Fig. 5e). By contrast, *sucla2*$^{-/-}$ zebrafish mutants that were raised on the transgenic background of *sirt5* overexpression showed significantly improved median survival, indicating that reducing the load of protein succinylation had benefited the health of the larvae (Fig. 5e). Since *sucla2* mutant animals show a reduced oxygen consumption rate and mitochondrial dysfunction commonly leads to a shift to anaerobic glycolytic metabolism, we also analyzed the survival of *sucla2* mutant and control animals in volume-restricted medium. In this set-up, zebrafish larvae were kept in 96-well plates at a defined volume of 200 μL where the metabolism of the animals leads to progressive acidification of the surrounding medium. We found that *sucla2*$^{-/-}$ animals died significantly earlier than wild-type controls. Transgenic over-expression of *sirt5*, by contrast, showed a trend to an improved median survival ($p = 0.146$) (Supplementary Fig. 7e).

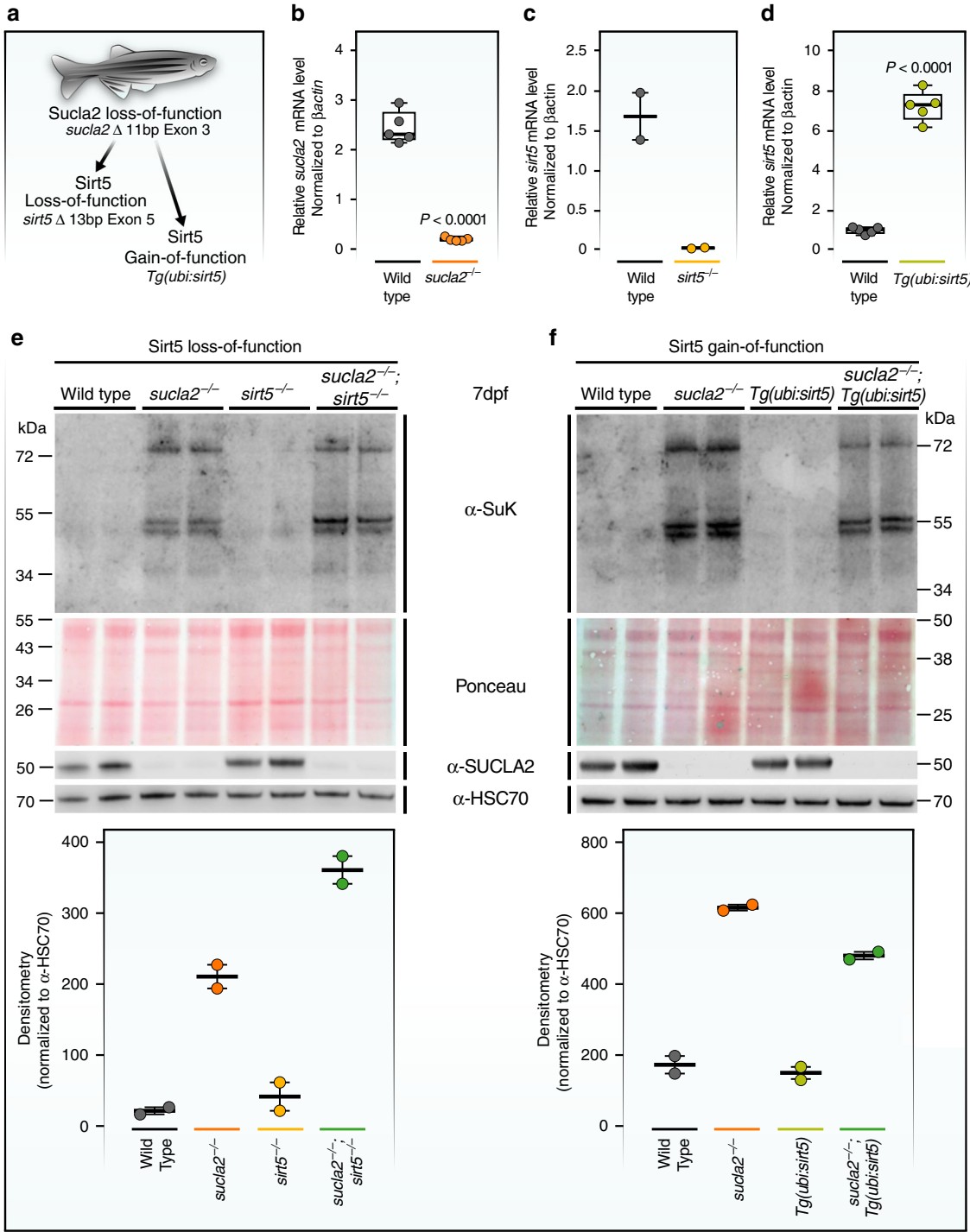

**Fig. 4 Sirt5 modulates global protein succinylation in a zebrafish model of SCL deficiency. a** Schematic representation of genetic backgrounds used for zebrafish experiments. **b** Relative *sucla2* mRNA levels in *sucla2⁻ᐟ⁻* zebrafish (*n* = 5 pools of 6 larvae, 7 dpf). **c** Relative *sirt5* mRNA levels in *sirt5⁻ᐟ⁻*zebrafish (*n* = 2 tissue samples from adult skeletal muscle, adults of 3 months). **d** Relative *sirt5* mRNA expression in *Tg(ubi:sirt5;cryaa:zsgreen)* transgenic zebrafish (*n* = 5 pools of 8 larvae, 8 dpf). All data in **b**–**d**, compared to wild-type siblings and normalized to *βactin* mRNA levels. *p*-values are calculated by standard unpaired *t*-tests for **b** and **d**. **e** SDS-page and western blot analysis of global protein succinylation in zebrafish larvae with homozygous mutations in *sucla2*, *sirt5*, or combined gene deficiencies and controls (7 dpf, *n* = 2 pools of 10–15 larvae) and **f** Pan-succinyl-lysine western blots of samples from larvae overexpressing *sirt5* in wild-type or *sucla2⁻ᐟ⁻* animals (*n* = 2 pools of 10 to15 larvae, 7dpf). Pan-succinyl-lysine antibodies were used to quantify lysine succinylation, and Hsc70 antibodies were used as loading controls. Ponceau staining serves as an additional loading control. Boxplots in **e**, **f** show quantifications of western blots using the mean of the densitometry of the three main bands, normalized to Hsc70. The boxplots show the median, the first to third quartile, and minima and maxima. Western blot images are derived from a single experiments. Source data are provided as a Source Data file. Bp = base pairs.

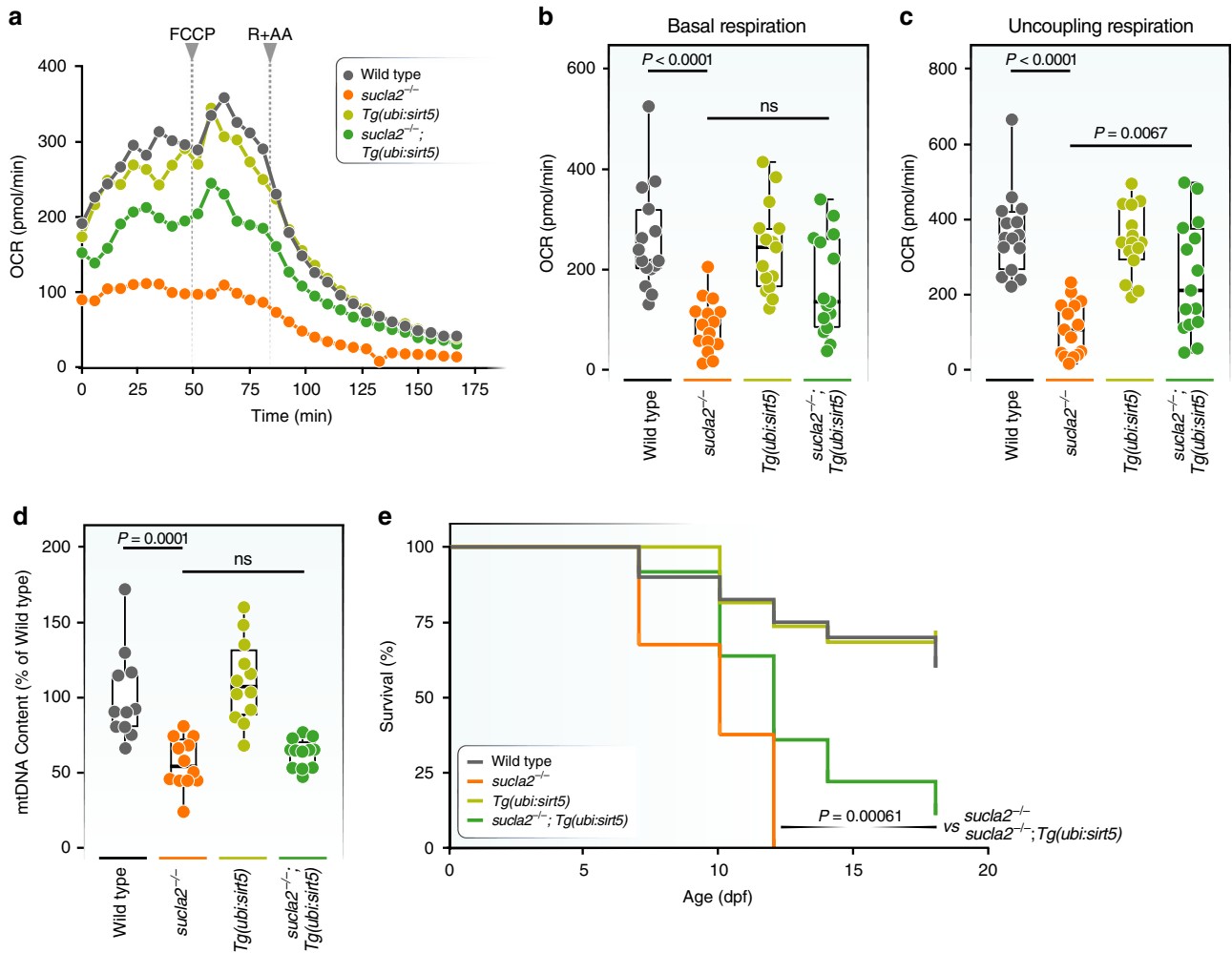

**Fig. 5 Sirt5 restores oxidative metabolism and improves survival of *sucla2*$^{-/-}$ zebrafish. a** Oxygen consumption rate (OCR) in *sucla2*$^{-/-}$ and control zebrafish larvae with or without *sirt5* overexpression ($n = 15$ per group, 10dpf), before and after the addition of inhibitors of mitochondrial respiration. The uncoupler FCCP is added to stimulate maximal OCR. Rotenone and antimycin A (R + A) determine non-mitochondrial respiration. **b** Basal OCR calculating the average of the three last points before addition of FCCP. **c** maximal OCR. **d** qPCR analysis of mitochondrial DNA content in *sucla2*$^{-/-}$ and control zebrafish larvae with or without *sirt5* overexpression ($n = 12$ per group, 10dpf). **e** Survival of *sucla2*$^{-/-}$ and control zebrafish larvae with or without *sirt5* overexpression in standard (fed) conditions (*sucla2*$^{+/+}$, $n = 40$; *sucla2*$^{-/-}$, $n = 37$; *sucla2*$^{+/+}$;*Tg(ubi:sirt5)*, n = 38; *sucla2*$^{-/-}$;*Tg(ubi:sirt5)*, n = 36). Data are pooled from two independent experiments. *p*-Value for median survival of wild type vs sucla2$^{-/-}$ <0.0001, p-value sucla2$^{-/-}$ vs sucla2$^{-/-}$;*Tg(ubi:sirt5)* = 0.00061. Pairwise comparison using log-rank test. *p*-Value adjustment was performed by Bonferroni correction. The boxplots show the median, the first to third quartile, and minima and maxima). n.s., not significant. mtDNA = mitochondrial DNA; dpf = days post fertilization. Source data are provided as a Source Data file.

We showed that protein succinylation in response to reduced SCL flux impacts on cellular function and pathological progression in vivo and that a targeted reduction of succinylation levels is a promising strategy to improve oxidative metabolism and health in SCL deficiency.

## Discussion

Mutations in *SUCLA2* or *SUCLG1* cause an incurable, progressive childhood-onset mitochondrial disease characterized by a progressive encephalomyopathy[22,35]. We used an untargeted approach to examine the metabolic consequences of SCL deficiency in patients and uncovered a potential PTM-linked disease mechanism. Our results show that SCL deficiency and the reduced flux through SCL lead to succinyl-CoA accumulation and widespread lysine succinylation of proteins in mitochondria and other cellular compartments. This finding is relevant because increased protein acylation is commonly associated with loss-of-function and changes in metabolic network regulation[6,11,36]. The

interactions of diet as source of acyl-CoA species with the genotype could lead to a variable degree of protein acylation and protein network dysfunction, and explain, at least partially, the heterogeneity in onset and severity of symptoms. Our proteomics approach identified almost 1,000 lysine residues, of which 899 are statistically increased in cell lines of patients with disease-causing *SUCLA2* mutations. These data suggest that, while baseline levels of succinylation are low, an increase of succinyl-CoA in mitochondria is sufficient to overcome the capacity of SIRT5 to efficiently remove succinylation marks.

The majority of succinylated proteins we identified are located within the mitochondria and affect bioenergetic pathways, including the TCA cycle, ATP-synthesis, and β-oxidation. Less expected was the observation of hyper-succinylation of proteins in pathways outside mitochondria, a phenomenon that has been described in other studies[9,26,30]; proteins responsible for cell-cell adhesion, branched-chain amino acid catabolism and glycolysis were also enriched for succinyl-lysine marks. In SCL deficiency,

the driver of succinylation is the intra-mitochondrial built-up of succinyl-CoA due to mutations in *SUCLA2* and, therefore, reduced enzymatic flux. Because succinyl-CoA is a negatively charged molecule and should not cross the mitochondrial membrane, the mechanism of lysine hyper-succinylation outside the mitochondria is not clear. Although pathways for transfers of acyl-groups across the mitochondrial membrane are known for other acyl-CoA moieties, no such mechanism has been reported for succinyl-CoA[37,38]. Cytosolic succinylation in this and other contexts could be caused by leakage of succinyl-CoA through damaged mitochondrial membranes, an unidentified transport mechanism for activated succinyl groups such as succinyl-carnitine, and secondary SIRT5 deficiency due to depletion of $NAD^+$ levels, the latter of which has been described in other forms of mitochondrial dysfunction or disease[13,39,40].

The greater average change of succinylation in response to loss of SUCLA2, compared to loss of SIRT5, was hinted at by the cross-species/cross-method comparison and substantiated in the direct comparison of loss-of-function genotypes in zebrafish. These observations suggest that the increase is substantial. Because these modifications converge on the same target proteins, these changes may cause pathological events that resemble those reported in *Sirt5*$^{-/-}$ mice[21]. For example, HADHA, a subunit of the trifunctional protein complex, was a major target with 27 detected lysine sites. Accumulation of succinylated HADHA could lead progressively to reduced β-oxidation, built-up of acyl-carnitines, and ultimately contribute to cardiomyopathy, a pathology that has been reported in cases of SUCLG1 deficiency. Identification of the target proteins is biased towards those of high abundance[30]. Nevertheless, the pathways consistently affected across studies represent important metabolic nodes for cellular energy metabolism and are likely to be functionally relevant.

Another important target that we identified was SHMT2, which was among the proteins with the highest numbers of succinyl-lysine sites. SHMT2 is a vitamin B6-dependent mitochondrial enzyme that fluxes serine into the one-carbon pool to derive glycine and methylenetetrahydrofolate. SHMT2 is involved in the synthesis of purines and controls mitochondrial translation initiation, linking SHMT2 activity with mtDNA maintenance and respiratory function[41,42]. Succinylation may, therefore, affect SHMT2's function by different mechanisms. Degradation of SHMT2 by targeted ubiquitination is mediated by acetylation of lysine 95 (ref. [43]). Furthermore, hyper-succinylation on lysine 280 reduces the enzymatic activity of SHMT2 (ref. [44]), although K95 and K280 were not among the eight succinyl-lysine sites found in our data set. However, our data show that, at least in the conditions studied here, the effects of SIRT5 on mitochondrial respiration are independent of changes in mtDNA content.

To test the functional impact of SIRT5 activation, we conducted a series of loss- and gain-of-function experiments in zebrafish as an experimental model that mimics the human disease. First, we showed that this zebrafish model recapitulates the phenotype of protein hyper-succinylation in the cells from SCL patients. The increase of protein succinylation in zebrafish appeared to be larger in response to *sucla2* deficiency than with loss of *sirt5*, confirming what was predicted by comparing data from patient-derived fibroblasts and *Sirt5*$^{-/-}$ mouse embryonic fibroblasts. Additional loss of *sirt5* further increased protein succinylation, and overexpression of *sirt5* could partially rescue elevated lysine succinylation, arguing that increases of succinyl-CoA override intrinsic SIRT5 activity in a dose-dependent manner. Interestingly, even zebrafish heterozygous for *sucla2* mutations accumulated succinylated proteins in skeletal muscle, an effect that was further increased in the background of *sirt5* loss-of-function. This finding is of interest as it suggests that protein succinylation occurs in recessive carriers of the disease, such as the children's parents, and that it may reduce mitochondrial energy metabolism with age or in other conditions associated with $NAD^+$ depletion. Consistent with the concept that *sirt5* is a disease modifier of SCL deficiency, zebrafish mutant larvae with extra copies of *sirt5* showed improved survival. We found that *sucla2*$^{-/-}$ mutant zebrafish died prematurely starting at 10 dpf. By contrast, overexpression of a *sirt5* transgene significantly increased survival. Of note, mtDNA levels were not rescued by increased Sirt5 protein levels, suggesting that mtDNA depletion is caused by mechanisms other than pathological succinylation. Metabolites directly upstream of SCL, including succinyl-CoA and α-ketoglutarate, were also higher than in wild-type animals suggesting that Sirt5 activation does not improve flux through SCL. The improved survival could be due to different effects of Sirt5 that mitigate the consequences of TCA cycle dysfunction in *sucla2* mutant zebrafish. These mechanisms could include in principle effects on any of the pathways identified in this study inside and outside the mitochondria, and desuccinylation activity of Sirt5 is likely to have a compound effect. Nevertheless, the partial restoration of oxygen consumption in zebrafish larvae overexpressing *sirt5* indicate that a rewiring from glycolytic to oxidative metabolism is at least in part responsible for the improvement in larval health. Future studies are warranted to dissect which energy fuels in particular allow a restoration of oxidative metabolism when the succinylation load is lowered in SCL deficiency.

Notably, although we identified succinylation as a modifier of SCL disease, the underlying genetic defect within the TCA cycle is likely the dominant pathomechanism responsible for the severity of the clinical symptoms. The concept of carbon stress acting on top of the genetic defect may explain the occurrence of clinical symptoms at a young age in patients in contrast to the age-related phenotypes observed in *Sirt5* knockout mice[21].

In sum, we found hundreds of proteins that are affected by protein succinylation in patients suffering from SCL deficiency. We provided evidence that pathological succinylation contributes to the disease pathology and may explain the heterogeneity of clinical symptoms in SCL deficiency. Looking forward, this study warrants exploration of strategies to increase sirtuin activity to slow SCL disease progression. These may include treatment with $NAD^+$ precursors, such as vitamin B3-forms (niacin, nicotinamide riboside, nicotinamide mononucleotide). Nicotinamide riboside remarkably improved disease signs in a mouse model of adult-onset mitochondrial myopathy due to mutations in the mitochondrial replicative helicase Twinkle[39], and adults with the same disease benefited from high-dose niacin[40]. Whether such "metabolic modifier" interventions and/or specialized diets that decrease the carbon load from succinyl-CoA, would benefit SUCLA2-patients, remains to be studied.

We show that pathological succinylation acts as a disease modifier independent of changes in mtDNA copy number and levels of TCA cycle metabolites. This finding suggests a possible multi-target approach of addressing the primary SCL defect and activating SIRT5 activity, for example, through membrane-permeable succinate to bypass the failure of producing succinate in the TCA cycle[45]. Beyond the relevance for SCL deficiency, our study provides evidence that non-physiological increases in reactive carbon species contribute to cellular dysfunction and disease through PTMs.

## Methods

**Protocol approvals, registrations, and patient consents**. All patients diagnosed with disease-causing mutations in *SUCLA2* admitted to the participating clinics, and from whom appropriate samples were available, were included in the study ($N = 7$). All patient samples were taken and used for research with informed

consent from the parents taken by the participating clinical institutes. All samples were taken for diagnostic purposes, and samples from age-matched pediatric patients with other mitochondrial diseases were used as controls, as samples from healthy children are not available. The study was approved by the Ethics Committee for Paediatrics, Adolescents and Psychiatry of Helsinki University Hospital and by the participating clinical institutes. Use of the samples is restricted to the research entities listed in the protocol. Fibroblasts were available from all patients (P1–P7), and myoblasts from one patient (P1). Metabolomics was performed on all fibroblast lines, and proteomics on fibroblasts from one patient with each of the disease-causing mutations ($n = 3$), and the myoblast line ($n = 1$).

**Cell culture and lysate preparation**. Fibroblasts from control patients and patients with mutations in *SUCLA2*, immortalized by retroviral transduction of E6/E7 proteins of human papilloma virus, were grown to 80% confluence and then cultured in 0.5% serum for 5 days. Cells ($3.5–4.8 \times 10^7$) in active growth phase were harvested by scraping for metabolomics analysis and trypsin for sodium dodecyl sulfate-polyacryylamide gel electrophoresis (SDS-PAGE) and proteomics (~3 mg of soluble protein per sample), washed and frozen as pellets. Similarly, myoblasts from two control individuals and one *SUCLA2*-deficient patient were differentiated into myotubes before collection of myoblasts and myotubes. Metabolomics analyses were performed on all fibroblast lines of the seven patients and three control subjects.

Proteomics were performed using three cell lines from patient-derived fibroblasts and from two control patients. Proteomics experiments on myotubes were performed using one patient-derived myoblast cell line and two patient-derived myoblast cells lines. To enable analysis of proteomics data of the myotubes, we compared two technical replicates of the patient line to two control myotube samples. Two pellets were collected from individual cultures of each cell line.

**Antibodies and reagents**. A table with antibodies and reagents used in this study is available in Supplementary Table 1.

**Sample preparation for proteomic experiments**. Frozen cell pellets from control and patient-derived fibroblasts and myotubes were thawed in lysis buffer (8 M urea, 50 mM Tris, pH 7.5, 1x HALT protease inhibitor cocktail, 150 mM NaCl, 5 μM trichostatin A, and 5 mM nicotinamide) and sonicated on ice with three bursts for 10 s with a probe-tip sonicator. Lysates were clarified by spinning at 20,000$g$ for 15 min at room temperature, and the soluble protein was moved to a new tube. Protein concentrations were determined using the BCA assay. Protein disulfides were reduced by adding DTT to a final concentration of 4.5 mM for 30 min at 37 °C, and free thiols were capped after cooling to room temperature by adding 10 mM iodoacetamide and incubating proteins for 30 min in the dark. Samples were diluted fourfold with 50 mM Tris buffer, pH 7.5, and proteins were enzymatically hydrolyzed overnight by adding trypsin at a ratio of 1:50 (wt:wt) trypsin to substrate. The enzymatic hydrolysis was then quenched by the addition of formic acid to 1% final concentration by volume, and the solution was clarified by centrifugation at 2000$g$ for 10 min at room temperature. The supernatant containing tryptic peptides was desalted using Oasis HLB vacuum cartridges (Waters). A small aliquot of the desalted peptide elution from each column containing approximately 20 μg was separated from the major portion of the peptides to be used for separate analysis of any protein-level changes. Desalted peptides were dried completely using a vacuum centrifuge. Dried peptide powder was then resuspended in IAP buffer (50 mM MOPS, 10 mM phosphate, 50 mM NaCl, pH 7.2) by pipetting, and succinylated peptides were enriched using anti-succinyl lysine antibody conjugated to agarose beads (Cell Signaling Technologies, PTMScan # 13764), according to the manufacturer's instructions except that 10 μL of beads were used for each sample (1/4 of a single tube containing 40 μL of beads). Succinylated peptides eluted from the antibody-bead conjugates in 0.1% trifluoroacetic acid in water were directly desalted using homemade StageTips[46]. Desalted peptides were dried completely by vacuum centrifuge and stored at −80 °C until they were resuspended in 0.2% formic acid in water for nanoflow liquid chromatography-tandem mass spectrometry (nLC-MS/MS) analysis.

**Proteomic data collection with nLC-MS/MS and data analysis**. All MS data were collected using an Eksigent nLC system with a cHiPLC column system coupled to a SCIEX TripleTOF 6600 MS or TripleTOF 5600 system using Sciex Analyst software version 1.7.

Peptides were analyzed by nLC-MS/MS based on the following established workflow[28,29,47]: A detailed description of the workflow is provided in the Supplementary Methods. Briefly, non-enriched peptides from the protein lysate were analyzed by data-independent acquisition to quantify proteome changes. The enriched fraction of succinylated peptides was analyzed twice: once by data-dependent acquisition to yield qualitative data and a second time with data-independent acquisition to produce quantitative data. All chromatographic and mass spectrometric scan parameters are detailed in the Supplementary Methods. Proteome data from non-enriched peptides was quantified using Spectronaut[48] and the pan-human spectral library (Spectronaut pulsar Version 11.0.15038.12.33511)[49]. Succinylated peptides were identified and quantified using a modification of the PIQED workflow, an automated analyses of post-translational modifications and protein quantification based on data-independent acquisition[29]: First, succinylated

peptides were identified by searching a database by data-dependent acquisition against the human proteome using MaxQuant[50] with the default settings except that variable succinylation of lysine was allowed. Next, the adapted PIQED analysis started from a spectral library of succinylated peptides built from MaxQuant identifications, which were quantified in Skyline[51]. Relative quantities of proteins were used to correct changes in observed PTM abundance before statistical comparison of quantities between patient samples and controls with mapDIA[52]. Changes to succinylation sites were considered significant when the log2 change was >1 at Q < 0.05 when different conditions compared to each other. For direct comparison of succinyl-lysine sites in mice (from Park et al.[26]) and human (our study), pairwise sequence alignments were performed to identify those sites that show succinylation in both sequences.

**Gene ontology term enrichment analysis**. Compared to controls, 353 proteins were hyper-succinylated in patient samples were from fibroblast data and were used as input for GO term enrichment analysis with DAVID[53]. Enriched terms are from the GOTERM_BP_DIRECT group; and the p-value used is from the fisher exact test.

**Identification and quantification of metabolites**. Intracellular metabolites from cells were harvested following standard protocols using 80% methanol/water as solvent[54–56], and dry pellets were stored in −80 °C freezer until ready for LC-MS analysis. For acyl-CoA analysis, dry pellets were reconstituted into 30 μL of sample solvent (water containing 50 mM ammonium acetate), and 10 μL was injected into the LC-MS. For non-acyl-CoA polar metabolite analysis, pellets were reconstituted into 30 μL of sample solvent (water:methanol:acetonitrile, 2:1:1, v/v/v), and 3 μL was injected into the LC-MS.

Mi 3000 UHPLC (Dionex) was used for metabolite separation. For polar metabolite analysis, a hydrophilic interaction chromatography method (HILIC) with an Xbridge amide column (100 × 2.1mm i.d., 3.5 μm; Waters) was used, and detailed parameter settings were according to standard protocols[57]. For acyl-CoA analysis, a reversed phase liquid chromatography method employing a Luna C18 column (100 × 2.0 mm i.d.; Phenomenex) was used[54]. LC was coupled with Q Exactive Plus mass spectrometer (Thermo Scientific) for metabolite separation and subsequent detection. The Q Exactive Plus mass spectrometer was equipped with a HESI probe, and for non- acyl-CoA polar metabolite analysis, it was operated in the full-scan mode with positive/negative switching with the resolution set at 70 000 (at $m/z$ 200). When it was used for acyl-CoA analysis, it was operated in the positive ion mode with a full-scan range of 150–1000 ($m/z$). LC-MS data were analyzed using Sieve 2.0 (Thermo Scientific), and the integrated area under metabolite peak was used to compare the relative abundance of each metabolite in different samples in the same batch. Retention times for each metabolite are included in Supplementary Data 1 and can also be found in methods papers outlining the experimental procedures applied in this study[54–56].

Intracellular metabolites from zebrafish were harvested from pools of eight larvae that were snap-frozen in liquid nitrogen and kept at −80 °C before analysis. A cold biphasic extraction was employed[58]. In short, tissues were homogenized in 300 μL of −20 °C methanol/water (5:3) using 5-mm stainless steel beads in a tissue grinder (Qiagen TissueLyser II) for 1:50 min:sec at 20 Hz. A fully labeled $^{13}C$ yeast biomass (50 μL) were added to the homogenate as internal standard. Further, 500 μL of methanol/water (5/3, v/v) (−20 °C) and 500 μL of chloroform (−20 °C) were added; the samples were agitated for 10 minutes at 4 °C in a thermo-shaker (Thermomixer C, Eppendorf), followed by 10 minutes of centrifugation. After extraction, the upper polar phase was recovered. The recovered extract was dried overnight in a vacuum centrifuge at 4 °C and 5 mbar, and then stored at −80 °C, before analysis. Dried samples were reconstituted in 20 μL of 60% (v/v) acetonitrile/water, and the supernatants transferred into glass vials for LC-MS analysis. Five microliters were injected into a Vanquish UHPLC (Thermo Scientific), equipped with a hydrophilic liquid chromatography column (ZIC-pHILIC column, 100 ×2.1 mm, 5 μm, with a ZIC-philic guard column 20 × 2.1 mm, 5 μm, both from Merck Sequant). Separation was achieved by applying a linear solvent gradient in decreasing organic solvent (90–25%, 15.5 minutes) at 0.2 mL/min flow rate and 35 °C. Aqueous 10 mM ammonium acetate with 0.04% (v/v) ammonium hydroxide (A) and acetonitrile (B) were used as mobile phases. The eluting metabolites were analyzed on an Orbitrap Fusion Lumos mass spectrometer (Thermo Scientific) with a heated electrospray ionization (H-ESI) source. The mass of the metabolites was assessed with on-the-fly positive and negative mode switching at a resolution of 60,000 at an m/z of 200. The spray voltages were 3500 V and 3000 V for positive and negative modes, respectively. The sheath gas was 20 AU, and the auxiliary gas was kept 15 AU. The temperature of vaporizer was 280 °C, and the temperature of the ion transfer tube was 310 °C. Instrument control and peak integration were conducted with the Xcalibur 4.2.47 software (Thermo Scientific). Metabolites were identified according to their exact mass, and the signal intensities were normalized to a $^{13}C$ internal standard.

**SDS-PAGE and immunoblotting**. Total cellular protein was extracted from patient and control fibroblasts and myoblasts. Protein extracts (5 μg) were separated on 12% SDS-PAGE (Criterion™ TGX Stain-Free, Biorad, Hercules, CA, Cat. #567-8045). Western blots for protein succinylation were performed by standard

method[17]. Uncropped western blot images are shown in Supplementary Figs. 2, 3 and 8. Antibodies as well as working dilution are specified in Supplementary Table 1.

**Protein structure modeling**. Structural modeling was performed using SWISS-Model Server and multiple sequence alignment using PROMALS3D Server. Model and Figure preparations were done with Discovery Studio v4.1 (Accelrys) as reported[24].

**Zebrafish husbandry and generation of zebrafish lines**. Adult zebrafish of the commonly used AB line were raised at 28 °C under standard husbandry conditions. All experimental procedures were carried out according to the Swiss and EU ethical guidelines and were approved by the animal experimentation ethical committee of Canton of Vaud (permits VD-H13 and VD3177). Transgenic zebrafish were generated using I-SCEI meganuclease-mediated insertion into AB embryos at the one-cell stage of a construct harboring the zebrafish *sirt5* under the control of the skeletal muscle-specific *actc1b* promoter or the ubiquitous *ubi* promoter. For rapid selection of transgenic animals, the injected constructs carried an eye-marker cassette harboring *ZsGreen* under the control of the *cryaa* (alpha-crystallin A chain) promoter in reverse direction. Transgenic carriers were outcrossed with AB fish to raise transgenic and wild-type siblings. The novel transgenic lines were registered at the central repository ZFIN.org under the designation *Tg(ubi:sirt5; cryaa:zsgreen1)*[nei005] and *Tg(actc1b:sirt5;cryaa:zsgreen1)*[nei006]. The transgenic lines can be found on www.zfin.org in the section "Tg/Mutants" entering the identifier (e.g. nei005) in the search field.

Gene disruptions of the *sirt5* and *sucla2* loci in zebrafish were generated using CRISPR/Cas9 genome editing. In brief, zebrafish embryos were injected at the one-cell stage with recombinant CAS9 protein and single guide RNAs targeting exon 3 of *sucla2* and exon 5 of *sirt5*. Primers were designed to amplify a short amplicon flanking the targeted sites, and high-resolution melt (HRM) analysis was performed on genomic DNA of the injected embryos to confirm successful gene disruption using a SYBR PCR mix on a LightCycler (Roche Life Science). Primer pair sequences were used as follows: sucla2: 5′- CTTGGTTATAAAAGCCCAAG TGC-3′ (forward) and 5′-GAGTAAACGATTCTGACTCCTCC-3′ (reverse); sirt5: 5′-GGGTGGGTAATTGGGAAGTT-3′ (forward) and 5′- GATGGTCCAGTCCT GGTTTG-3′ (reverse). Adult F0 zebrafish were outcrossed to wild-type AB animals to generate a F1 generation. HRM analysis on genomic DNA from tails of zebrafish adults was used to identify founder animals, and the out-of-frame mutations were confirmed using gene sequencing. The zebrafish mutant lines were registered at the central repository ZFIN.org with the designations *sucla2*[−/−nei010] and *sirt5*[−/−nei004]. The mutant lines can be found on www.zfin.org in the section "Tg/Mutants" entering the identifier (e.g., nei004) in the search field.

**Gene expression and mtDNA quantification**. For gene expression analysis, flash frozen pooled larvae or adult muscle were lysed in Qiazol with the FastPrep®-24 tissue homogenizer (MP-Biomedials). Total mRNA was extracted using the QIA-cube plateform and mRNAeasy kit (Qiagen). For RT-qPCR, cDNA was synthesized using standard cDNA synthesis kit, following the manufacturer's instructions (Thermo Fisher). qPCR was performed with the Roche Light Cycler 480 using SYBR green kit (Maxima SYBR Green, Thermo Fisher) and the following primers: βactin-Fwd: GTGGTCTCGTGGATACCGCAA, βactin-Rev: CTATGAGCTGCCT GACGGTCA, sirt5-Fwd: GGGAGTTTTACCATTACAGGCG, sirt5-Rev: ACATG TTTAGACCCAGCCCG, sucla2-Fwd: CCGCTGGACTGAGGAATTCC, sucla2-Rev: TCATGCAGGGAGAGCTTTCT

For mtDNA quantification, genomic DNA from zebrafish larvae of the four genotypes *sucla2*[+/+], *sucla2*[−/−], *sucla2*[+/+];*Tg(ubi:sirt5)*, and *sucla2*[−/−];*Tg(ubi: sirt5)* was extracted and precipitated with ethanol. MtDNA content was determined by qPCR as above, using *mt-co1* as the mitochondrial gene target and *polg1* gene as a reference for nuclear DNA content. The following primers were used: polg1-Fwd: GAGAGCGTCTATAAGGAGTAC, polg1-Rev: GAGCTCATCA GAAACAGGACT, mtco1-Fwd: ACTTAGCCAACCAGGAGCAC, mtco1-Rev: TC GGGGAAATGCCATATCGG.

**Oxygen consumption rate measurements**. Zebrafish larvae of the four genotypes *sucla2*+/+, *sucla2*−/−, *sucla2*+/+;*Tg(ubi:sirt5)*, and *sucla2*−/−;*Tg(ubi:sirt5)* were placed in wells of a 24-well microplate (one larvae in 600 μL of egg water per well, *n* = 5 per group) and maintained with an islet capture screen. The plate was loaded into an XF24 Extracellular Flux Analyzer (Seahorse Bioscience), and OCR was measured every 5 min. Basal respiration was calculated on the average of the last three point measurements before the injection of 2 μM FCCP to permit the electron transport chain (ETC) to function at its maximal rate. Uncoupling respiration was determined by the maximal value after FCCP injection. Finally, 1 μM Rotenone and 1 μg/mL Antimycin A were injected to shut down the ETC and reveal the non-mitochondrial respiration, which was deducted from both basal and uncoupling respiration values. Temperature was kept at 28.5 °C during the whole experiment. Experiment was repeated three times in the exact same conditions to ensure reproducibility and solidity.

**Zebrafish survival assays**. Zebrafish larvae of the four genotypes *sucla2*[+/+], *sucla2*[−/−], *sucla2*[+/+];*Tg(ubi:sirt5)*, and *sucla2*[−/−];*Tg(ubi:sirt5)*, were transferred into 1.1-liter tanks and kept under constant water flow. The larvae were fed twice daily with dry food, and dead larvae were identified and removed daily, and the genotype was confirmed. Experiments in volume-restricted conditions were performed by transferring zebrafish larvae into 96-well plates in a volume of 200 μL. The health of the larvae was monitored frequently, and dead larvae or larvae with edema or abnormal heartbeat were removed, followed by confirmation of the genotype.

**Statistical analysis**. All statistical analysis was performed as indicated in the main text and figure legends. Longitudinal data of zebrafish larvae were analyzed with Kaplan-Meier survival curves with pairwise comparison using log-rank test. *p*-Value adjustment was performed by Bonferroni post-hoc test. Box-plot elements show the following data: center line, median; box limits, upper and lower quartiles; whiskers, 1.5x interquartile range; points, outliers.

**Reporting summary**. Further information on research design is available in the Nature Research Reporting Summary linked to this article.

## Data availability

Metabolomics data for d0 and 5 fibroblasts were deposited at Metabolomics Workbench with the project number PR000991 [https://doi.org/10.21228/M8M116].

Raw mass spectrometry data of protein hyper-succinylation in fibroblasts and myotubes derived from patients with mutations in succinyl-coA ligase along with supplemental tables of identifications and quantification are available from the UCSD proteomics resource Massive under the accession codes MSV000082599 [https://massive.ucsd.edu/ProteoSAFe/dataset.jsp?task=7ca3028f854e4996b58f1fa1fc2286fd] and PXD010392 [http://proteomecentral.proteomexchange.org/cgi/GetDataset?ID=PXD010392]. The Skyline document containing the PTM spectral library and quantitative data is available from Panorama at https://panoramaweb.org/project/Schilling/SuccinylCoALigase/begin.view?.

Proteomics data from Rardin et al.[17] used to determine changes in lysine succinylation of TCA cycle subunits in mouse liver can be found at https://www.cell.com/cms/10.1016/j.cmet.2013.11.013/attachment/74906873-3844-47b3-8cc5-a66cb052ecbf/mmc3.xlsx (protein quantification) and https://www.cell.com/cms/10.1016/j.cmet.2013.11.013/attachment/6c379a66-daf2-413f-884a-c23a478fcbae/mmc4.xlsx (peptide quantification)

The proteomics analysis from Park et al.[26] on changes in lysine succinylation in fibroblasts from control and *Sirt5*[−/−] mice can be found at https://www.cell.com/cms/10.1016/j.molcel.2013.06.001/attachment/59f72482-fad7-48fd-a9fa-9f38fd7a9232/mmc2.xls

The homology modeling of the human SCL heterodimer is based on the structure *E. coli* Succinyl-CoA synthetase with the PDB code 1CQI [https://doi.org/10.2210/pdb1CQI/pdb].

Source data are provided with this paper. All other experimental data that support the findings of this study are available from the corresponding authors upon reasonable request. Source data are provided with this paper.

## Code availability

R-scripts used in this study can be downloaded at https://github.com/jessegmeyerlab/SUCLA2-deficiency.

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

## Acknowledgements

The authors wish to thank Tuula Manninen, Babette Hollman, Anu Harju and Markus Innilä for technical assistance. We thank Thierry Guillaud, José Sanchez, Eduardo Jiminez, and Noélie Rochat for assistance in the zebrafish facility, John C. Carrol for graphic design and Gary Howard for editing the manuscript. The study was supported by grants from the Academy of Finland, Sigrid Jusélius Foundation, University of Helsinki and Helsinki University Hospital (to A.S.). This work was supported by the National Institute of Diabetes and Digestive and Kidney Diseases (R24 DK085610 E.V.). We acknowledge support from the NIH shared instrumentation grant for the TripleTOF system at the Buck Institute (1S10 OD016281, Buck Institute).

## Author contributions

P.G. and E.V. conceived the study. P.G., S.M., J.G.M, B.S., A.S. and E.V. contributed to the design of the study and wrote the manuscript with help of the other co-authors. J.G.M. performed the proteomics analysis. W.H., Y.N., and J.C.N. contributed to proteomics workflow. J.R. generated the zebrafish lines and performed zebrafish experiments. S.M., P.P. X.L. and S.C. performed metabolomics experiments and S.M. and J.W.L. supervised metabolomics analyses. P.P., C.J.C., L.E., C.B.J., G.C., A.P., J.T., and C.C. contributed to experiments of the study. P.I., B.A.M., R.A.A. and E. Ø. obtained and provided human fibroblast samples. All authors reviewed the manuscript.

## Competing interests

J.R., G.C., J.T., S.C., S.M. and P.G. are employees of Nestlé Research. No patents have been filed related to work conducted in this study. The remaining authors declare no competing interests.
