## [Peer Review File · Nature Communications]

Reviewers' comments:

Reviewer #1 (Remarks to the Author):

The authors are interested in acyl modifications of proteins on select lysyl residues. Here they focus on protein lysyl succinylation, first establishing a large set of proteins that undergo this post translational modification in mouse livers, with very nice comparisons of wildtype, and desuccinylase Sirt5-deficient samples (and parallel studies on protein acetylation in wildtype and deacetylase Sirt3-deficient samples). They also characterize human fibroblasts and myotubes from subjects with loss-of-function mutations of the succinyl-CoA ligase beta subunit *SUCLA2* (a Mendelian in-born error of metabolism called succinyl-CoA ligase deficiency). The authors then catalog the proteomic and succinylation changes that occur in *SUCLA2* deficiencies, and compare these findings to changes in Sirt5-null mouse fibroblasts. This analysis shows rather nicely that enzymes of succinate metabolism are among the most abundantly succinylated. Finally, the authors prepare and characterize a new zebrafish null allele of the orthologous gene *sucla2*. The zebrafish studies include a transgenic over-expression component with the desuccinylase SIRT5. They find that *SUCLA2* mutation causes global increase in protein succinylation, and renders larvae and early juveniles sensitive to death generally and under acidic medium conditions.

The study includes rigorous proteomic and metabolomic studies and presents novel tools for studying succinylation. The manuscript is cast as a pathogenesis mechanism study for a rare in-born error of metabolism; however, with some improvements, this study could have very broad implications for signaling and metabolism, generally.

1. **Mitochondrial respiratory complex assembly.**

Ref. 22 (among others) shows rather diverse changes in respiratory function among patients with succinyl-CoA ligase deficiency. The authors should state the activities of the primary cells they work with (both those obtained from previous publications and their newly reported sample). Similar studies in the larval zebrafish models: the somites of embryos and the subsequently mature (maturing) skeletal muscles are quite abundant and complexes I through IV activity should be easily measured.

2. **Interface of ketone utilization and succinylation.**

Although not rising to the degree of statistical significance, the rate-limiting enzyme of ketolysis (OXCT1 (3-oxoacid CoA transferase 1; formerly called succinyl CoA:3-oxoacid CoA transferase, SCOT)

is succinylated and differentially expressed in their patient samples. Acetoacetate is also increased (albeit not significantly, both d0 and d5) in fibroblasts. OXCT1 catalyzes the CoA-thioesterification of the ketone body acetoacetate, taking the CoA from succinyl-CoA (as the former name SCOT indicates; and not free CoA-SH) to activate this fuel for oxidation. Since cell culture in complete medium may not reveal changes in ketone utilization, the authors might find a critical pathogenetic mechanism-defective ketone utilization- by incubating cells in low-glucose medium and adding 3-hydroxybutyrate (easier) or acetoacetate (more challenging since sodium salts invariably decarboxylate spontaneously, and it is tricky to work with esters and anhydrides for fresh, slightly alkaline preparation and treatment of cells in culture).

3. **Zebrafish tools:**

a. the molecular lesion carried by the *sucla2* mutant should be presented formally. Stating what exon was targeted and even listing the guide RNA does not provide sufficient detail (and will be difficult to inventory in zfin.org). The full deletion (as a DNA sequence and a cartoon of the chromosome and the protein if a truncation is made), and description of the heterozygous carriers should be presented.

b. There is mention of a *sirt5*^{sup} null mutant in the methods, but no experiments are presented with this model.

c. Additional details regarding the Sirt5-transgenic should be presented: is this transgenic model used heterozygously (i.e., 1 integration site achieved after outcrosses)?

4. **Zebrafish survival analysis.**

a. Figure 4c should be presented in the standard format used in 4d (i.e., a step function, not curves with error bars-adding up cohorts of zebrafish undergoing survival analysis is acceptable, just as is rolling recruitment of human subjects for trials-not all subjects are recruited on the same day, nor are they the exact same age).

b. The log-rank test approach used is not standard. Even with adjustments with the post-hoc test selected (Benjamini-Hochberg), there is substantial risk of seeing differences that arise from chance. A simpler approach is to perform the conventional log-rank tests and then perform a Bonferroni correction. The key aspect is to be transparent about how many comparisons are being made: the K needs to be on the higher end, as explained clearly in this work flow for the very popular statistical software used by the authors (<https://www.graphpad.com/support/faq/after-doing-logrank-analysis-on-three-or-more-survival-curves-can-i-perform-multiple-tests-for-differences-between-pairs-of-curves/> - this product, incidentally, should be called 'Prism' in the first column of the Antibodies and reagents table; and the source is GraphPad (what they call "Prism GraphPad is analogous to calling a unique car model and make when the column calls for just a model).

c. Important controls for 4D are needed to firmly cement that acidification of the extracorporeal environment is the driver of death: in one experiment, adding 5 mM HEPES pH 7.2 (as in the classic

volume restriction-live imaging study: Westerfield M , Liu DW , Kimmel CB , Walker C. (1990). Pathfinding and synapse formation in a zebrafish mutant lacking functional acetylcholine

receptors . Neuron 4 , 867 – 874) will serve as a buffer against the acid of respiration and the acid from the excreted toxic metabolites (not just methylglutaconic acid, as ref. 22 describes-indeed, it is methylglutaconic acidemia that appears most specific for succinyl-CoA ligase deficiency in that large cohort study). A second control would involve changing the E3 medium over the course of the survival study, since this would remove toxic metabolites and acid that accumulate.

Reviewer #2 (Remarks to the Author):

This manuscript by Gut et al. focuses on the interplay between lysine succinylation (suK), the mitochondrial disease succinyl-CoA ligase (SUCL) deficiency, and the lysine deacylase SIRT5. The authors find that accumulation of succinyl-CoA in context of SUCL deficiency leads to hypersuccinylation of many mitochondrial and non-mitochondrial proteins; these suK sites largely overlap with those occurring in SIRT5 KO cells. The authors generate a fish model of SUCL deficiency and show that global hypersuccinylation, and early mortality, are partially rescued in this strain by concomitant SIRT5 overexpression.

Overall, this manuscript is well-conceived and -executed, and makes a significant novel contribution to the study of lysine post-translational modifications. Moreover, it may have significant downstream therapeutic implications, given that SUCL deficiency, although rare, is a severe mitochondrial disease with no effective therapy available currently. I recommend publication in Nature Communications, provided that the authors can address the following relatively minor concerns:

1. Given that the authors make conclusions about relative global suK levels between samples based on immunoblots, these blots (Figs. 1e, 1f, 4a, and 4b) need to be quantified.
2. According to the legend to Fig. 1, "Controls (C1-C3) are fibroblasts from age-matched patients with other mitochondrial diseases". The authors should also compare suK levels in a couple of truly wild-type (not mutant) fibroblast lines to SUCL-deficient cells.
3. I think caution is in order when comparing the magnitude of hypersuccinylation in SIRT5-deficient mouse embryo fibroblasts compared to human SUCL-deficient cells (p. 12). Such differences may stem from species differences, differences in culture conditions, differences in cell type (embryo-

derived fibroblasts versus adult skin fibroblasts), etc. These caveats should be mentioned by the authors.

4. Overexpression of SIRT5 in the transgenic fish line should be documented, preferably by western blot.

5. In Fig. S2, the authors have inadvertently omitted the designation of SIRT5 overexpression.

6. One interesting question that arises from these studies is why the phenotypic effects of SIRT5 deficiency are comparatively mild, whereas SUCL deficiency is a severe disorder. This may warrant attention in the Discussion section.

7. Although I do not feel that the absence of this study is a deal-breaker for this manuscript, it would have been great to generate a SIRT5HY catalytic null transgenic fish line, to test directly whether SIRT5 activity is required for the rescue observed.

Reviewer #3 (Remarks to the Author):

The manuscript by Gut et al (SUCLA2 mutations cause global protein succinylation contributing to the pathomechanism of a hereditary mitochondrial disease) provides evidence that succinyl-CoA ligase (SCL, SUCLA2) deficiency leads to excess accumulation of succinyl-CoA, and that the resultant increase in protein lysine succinylation contributes to the disease phenotype of individuals harboring mutations in SUCLA2.

The authors compare previously published datasets of protein acetylation and succinylation in mouse liver tissue, and use quantitative mass spectrometry to analyze succinyl-CoA and succinylation levels in fibroblast and myotube cell lines from SUCLA2-deficient patients and from control individuals. Bioinformatic analyses are performed to compare increased succinylation in SCL and a previous dataset of SIRT5-regulated sites. In the penultimate experiment, the authors show that SIRT5 overexpression can suppress the lethality SUCLA2 mutant zebrafish.

While the authors provide intriguing evidence that pathological succinylation caused by increased succinyl-CoA may contribute to disease in individuals with SUCLA2 mutations, the bioinformatics analyses of protein succinylation are flawed and the data is sometimes presented in a misleading manner. The analyses of SCL deficient zebrafish is well done and supports their hypothesis, but could be better supported with additional experiments. In particular, the manuscript would be greatly improved by avoiding flawed bioinformatics analyses that are over-reliant on comparing numbers of identified sites and instead quantify how SIRT5 and SCL deficiency interact at the site level in the model systems (cell lines and zebrafish) employed in this study.

Major comments:

1. Why is the analysis of previously published acetylation data included in this manuscript? The authors claim that these data show that TCA enzymes are particularly susceptible to nonenzymatic acetylation. However, there is no data presented here or elsewhere to suggest enrichment of SIRT3-regulated sites on TCA proteins compared to other mitochondrial proteins. The analysis in Supplementary Figure 1A is flawed because it only compares the number of identified sites on TCA proteins compared to all proteins. Several studies show that more acetylation (and succinylation) sites are identified on abundant proteins (PMID: 30837475, and Ref#11) due a technical bias to identify acylated peptides from abundant proteins. Since TCA proteins are highly abundant this result is completely unsurprising, are TCA proteins enriched when compared to similarly abundant mitochondrial proteins? To perform this analysis correctly you should also correct for the lysine content of the analyzed proteins, large proteins with numerous lysines will harbor more sites than smaller proteins with fewer lysines. See also (PMID: 26513550) for gene ontology analysis that is corrected for protein abundance bias.

How does including analysis of acetylation sites contribute to this manuscript or our understanding of the role of succinylation in SCL pathology.

The analysis of succinylation in Supplementary Figure 1A is similarly flawed and should be corrected or removed.

I suggest that the authors use abundance corrected intensity (acetylated peptide intensity corrected for protein abundance by dividing by total protein intensity) to determine relative acetylation stoichiometry between different classes of proteins (for example TCA proteins versus other mitochondrial proteins, or SCL proteins versus other mitochondrial proteins). This measure is independent of protein abundance bias and can suggest higher levels of acetylation within certain classes of proteins, we first described this in yeast (PMID: 24489116) and recently used it to compare different classes of CBP/p300-regulated acetylation sites (PMID: 29804834).

2. Page 8, 164-166. The authors state “Notably, of all enzymes of the TCA cycle, SCL subunits had by far the largest fold change among Sirt5-regulated acylation sites: three subunits, Suclg1, Sucla2 and Suclg2, had several Sirt5-regulated lysine residues each (Fig. 1b and Supplementary Fig. 1c,d).”

There are several problems here. The authors refer earlier in the manuscript to 3 studies from which these data are derived, 2 of which were studies of succinylation, Rardin et al, and Park et al. First, it is unclear which of these two studies the data shown in Supplementary Figure 1c,d comes from. Most importantly, the claim that these sites have by far the largest fold change is false. By comparing the data shown in Suppl. Fig. 1d to the data from each of these studies it is clear these

data come from the Rardin et al study. However, the largest change shown in Suppl. Fig. 1d is 61.6 fold for Suclg1 K94, in the data from Rardin et al, this is the site with the 12th highest increase, Atp5o is top with a 395-fold increase. The second highest site shown in Suppl. Fig. 1d is Sucgl1 K66 with 25.6, which is the 32nd most increased site in their data. The data clearly does not support the claim that SCL enzymes have “by far the largest fold changes” and I assume that the problem here is a mistake in describing these data, because as currently described, it is not true, at all. Can the authors support this claim by using an actual statistical test, (for example, comparing the maximum, median, and/or average fold changes between regulated proteins), and using data from both studies?

3. The authors claim that sites on suclg1 and sucla2 occur within the structure of these proteins at positions that are likely to impact their enzymatic function. However, the stoichiometry (degree of modification) at nonenzymatically acetylated lysine residues has been shown to be extremely low. Since the manuscript hypothesizes that nonenzymatic succinylation drives (some of) the disease phenotypes, can the authors please determine the stoichiometry of modification using peptide standards, either in SIRT5 and/or SCL deficient cells or tissues. These measurements would be important to support the claim that “These data suggest that the remaining SCL activity in mutations leading to hypomorphic phenotypes can be further reduced by succinyl-CoA-mediated auto-succinylation.”

4. Fig 2g and Page 10 215-219

It is incorrect to compare a group of modified proteins to all proteins in gene ontology analysis as shown by (PMID: 26513550). This analysis only shows the abundance bias of succinylation detection. Furthermore, since the process is nonenzymatic we expect all proteins to be succinylated, which is consistent with the observation that all (98%) of sites were increased in patient cells. Why then perform gene ontology analysis if all proteins are affected? (but you only detect some of them because they are more abundant). It would be more informative to compare the group of proteins showing the greatest increases in succinylation to those showing the least increase in succinylation (if possible since this likely varies at the site-level and not the protein-level).

5. Fig 2h

The figure sorts proteins based on the number of detected succinylation sites. As stated above in major comment #1, this analysis is compounded by the technical bias to identify a greater number of sites on abundant proteins and is further compounded by the number of lysine residues on those proteins and whether they can be detected as tryptic peptides in the MS. Furthermore, we expect that all solvent accessible lysines are nonenzymatically succinylated, so there is no point in this analysis. The analysis is flawed.

p11, 223-225

“In summary, succinylation in response to high levels of reactive succinyl-CoA affects proteins of diverse cellular pathways, and the TCA cycle and metabolic pathways are major targets.”

The TCA and metabolic pathways are not necessarily major targets, it is just easier to identify succinylation sites on these abundant proteins.

6. Fig3A

The presented overlap in observed proteins between SUCLA2 deficiency and SIRT5 deficiency is misleading. While many of the same proteins were identified in these two studies, only 124 sites on 48 proteins were found in both studies. This explains why the authors fudge the following analyses (Fig3B-E) by comparing the numbers of succinylation sites found on proteins instead of comparing the actual changes occurring at the same sites in both experiments. The site-level overlap for these studies is only 13% for SCL sites, compared to the 51% they show for proteins in the figure, this is misleading.

7. Fig3C and D

As stated above several times, several studies have shown that we identify more sites per protein on abundant proteins (the authors are welcome to determine protein abundance in their own samples, which is easy, and verify this bias in their own data). That the authors identified a large number of sites on the same proteins as in a study of SIRT5 deficiency is both unsurprising and uninformative. What is the significance of this finding? The same relationship would almost certainly occur if you compared normal fibroblasts from mice to fibroblast from human. This type of control analysis is lacking, and a technical bias is overinterpreted. Sadly, there is plenty in this manuscript that is interesting without relying on technically flawed analyses such as this.

8. Fig3B and E

It is an interesting finding that SCL patients show more substantially increased succinylation at a majority of sites compared to SIRT5 deficiency (although SIRT5 deficiency causes more dramatic increases at a subset of sites). However, the authors miss a chance to investigate how SIRT5-regulated sites are impacted in SCL. Are SIRT5-regulated sites increased in SCL to the same degree as all sites? In other words, does SIRT5 suppress increased succinylation in SCL? This is particularly relevant question given the results shown in the Zebrafish model. However, I assume that the authors are unable to address the question since only 124 sites are quantified in both studies, thus the authors are comparing different sites when performing this analysis. It would be best if the

authors KO'd SIRT5 in this SCL cell lines to directly investigate the interaction between SIRT5 and SCL deficiency.

In Fig3E, please indicate sites that were common between the studies by connecting them with lines, this would indicate the trend between SIRT5 regulation and SCL impact at individual sites.

9. P12, 256-260

“These data show an extensive overlap between proteins that accumulate lysine succinylation marks in response to SUCLA2 and SIRT5 deficiencies and suggest non-random succinylation events independent of whether succinylation is derived from increased succinyl-CoA levels or decreased lysine de-succinylation activity.”

This statement is not supported by the data and is misleading.

1. All proteins (98%) show increased succinylation in SCL, which is expected based on the nonenzymatic mechanism, so there is no special significance that these sites overlap with SIRT5 regulated sites (if all sites are regulated they will overlap with any sites found in any study).
2. The only thing that is non-random is that the authors employed antibody enrichment of succinylated peptides that is similarly biased for peptides from abundant proteins. To show that this is indeed non-random the authors would need to correct for protein abundance biases, or perform a similar comparison of patient derived normal human fibroblasts to normal mouse fibroblasts.
3. Since the analysis relies on comparing the number of sites found on each protein it says nothing about the relationship between degree of SIRT5 regulation and degree of succinylation in SCL. This would be much more informative.

10. Fig4C and D

Why were *sucla*^{-/-}, *sirt5*^{-/-} animals not analyzed? If SIRT5 protects against increased succinylation we would predict that these double mutant animals would show decreased survival compared to *sucla*^{-/-} animals. The data for these experiments should also be shown and the results discussed.

Minor comments:

1. Page 4.

“This prediction was further validated experimentally in vitro 4-7.”

The authors should at least cite (PMID: 23946487) and (11) which were the first studies to show this in vitro and were published at about the same time.

2. Page 5, 106-107

“Research with yeast indicates that disrupting the succinyl-CoA ligase reaction increases global protein succinylation 21.”

The citation is incorrect, the cited paper did not investigate yeast. The citation you want is (11).

3. p11, 237-240

“The similar outcomes of these different analyses suggest specificity for a subset of cellular proteins independently of the succinylation driver (i.e., both the loss of SIRT5 and elevated succinyl-CoA levels, as shown in the present study) converge on a similar set of target proteins.”

A major premise of this paper is that succinylation is nonenzymatic. What is the basis of the proposed specificity? Proximity? Unfortunately, none of the included bioinformatic analyses showed this. Also, since all sites are impacted by SCL, they will converge on any group of succinylation sites that you compare these data to.

4. p13, 280-282

“Combined deficiency of sirt5 and sucla2 further increased global protein succinylation, suggesting that succinylation is driven in vivo both by succinyl-CoA and by sirt5 (Fig. 4a).”

Can you provide quantification? The Western blot images are not particularly convincing.

5. p16, 344-355

“The majority of succinylated proteins we identified are located within the mitochondria

and affect bioenergetic pathways, including the TCA cycle, ATP-synthesis, and beta-oxidation. Less expected was the observation of hyper-succinylation on proteins in pathways outside mitochondria. Proteins responsible for cell-cell adhesion, branched-chain amino acid catabolism and glycolysis were also found to be enriched for succinyl-lysine marks. The sources of these lysine-succinylation reactions are not clear, although increased levels of succinyl-lysine modifications have been described in conditions of SIRT5 loss-of-function 21. In SCL deficiency, the driver of succinylation is the built-up of succinyl-CoA due to mutations in SUCLA2 and, therefore, reduced enzymatic flux. Cytosolic succinylation in this context could be caused by a leakage of succinyl-CoA through damaged mitochondrial membranes or secondary SIRT5 deficiency due to depletion of NAD⁺ levels, which has been described in other forms of mitochondrial dysfunction or disease 12,37.”

Although succinylation is biased to mitochondrial proteins in human cells and mouse liver (11 and PMID: 26513550), succinylation sites are found outside of mitochondria in the absence of pathologies such as SCL or SIRT5 deficiency (11, 21 and PMID: 26513550), therefore the authors should avoid using pathological explanations for this observation.

Reviewed by Brian Weinert

Reviewer #4 (Remarks to the Author):

The authors reported on their study hyper-succinylation due to SUCLA2 deficiency with investigations into countering effects of sirtuins. This is an important area of research to improve our understanding how protein acylation affects key cellular energy pathways. The authors highlighted the elevated levels of protein succinylation and buildup of succinyl-CoA, while only moderate recovery from elevated level was observed in sirtuins expression. The authors did an excellent job in reporting their finding as the manuscript was easy to read and understand. Here are a few revisions and suggestions:

-Lines 13-27, punctuation inconsistencies within the affiliations.

-Line 458, “Formic Acid” does not need capitalization.

-Lines 493-494, remove “and detection”.

-Line 487, Ref 47 described six different extraction methods, need to indicate extraction solvent(s) used.

-Line 509, how much standard was added?

-Line 517, "+ focused liquid chromatography system" is not needed.

-Metabolite supplemental, include the more information concerning the annotation. The annotation based only on accurate mass and the accurate mass of the ^{13}C signal from the Yeast biomass. Add the exp m/z (analyte and standard) and RT to the spreadsheet. The chirality is not usually determined, so that should be removed from the names. Some of the formulas reported don't seem likely candidates.

-Similar to the authors sharing of the proteomic data, the metabolomics data could also be loaded into a public repository like Metabolomics Workbench.

Reviewers' comments:

Reviewer #1 (Remarks to the Author):

The authors are interested in acyl modifications of proteins on select lysyl residues. Here they focus on protein lysyl succinylation, first establishing a large set of proteins that undergo this post translational modification in mouse livers, with very nice comparisons of wildtype, and desuccinylase Sirt5-deficient samples (and parallel studies on protein acetylation in wildtype and deacetylase Sirt3-deficient samples). They also characterize human fibroblasts and myotubes from subjects with loss-of-function mutations of the succinyl-CoA ligase beta subunit *SUCLA2* (a Mendelian in-born error of metabolism called succinyl-CoA ligase deficiency). The authors then catalog the proteomic and succinylation changes that occur in *SUCLA2* deficiencies, and compare these findings to changes in Sirt5-null mouse fibroblasts. This analysis shows rather nicely that enzymes of succinate metabolism are among the most abundantly succinylated. Finally, the authors prepare and characterize a new zebrafish null allele of the orthologous gene *sucla2*. The zebrafish studies include a transgenic over-expression component with the desuccinylase SIRT5. They find that *SUCLA2* mutation causes global increase in protein succinylation, and renders larvae and early juveniles sensitive to death generally and under acidic medium conditions.

The study includes rigorous proteomic and metabolomic studies and presents novel tools for studying succinylation. The manuscript is cast as a pathogenesis mechanism study for a rare in-born error of metabolism; however, with some improvements, this study could have very broad implications for signaling and metabolism, generally.

We thank the reviewer for the thorough evaluation of our study and for pointing out the strengths of the findings describing a new pathomechanism in SCL deficiency. We appreciate that the reviewer points out that we generated a set of new tools for this study. First, these include a catalogue of succinylation PTMs on the cellular proteome of patient and control fibroblasts. Second, we contribute a set of zebrafish models to study the interplay of SCL deficiency and sirtuin-dependent regulation of lysine succinylation. As next steps, these tools can be used to deepen our understanding of SCL disease and cellular signaling by reactive carbon species as pointed out by the reviewer. However, to keep the study focused and also due to the restrictions to lab access during the COVID-19 crisis, we could not consider further wet-lab experiments to understand succinyl-lysine signaling in a broader context.

1. Mitochondrial respiratory complex assembly.

Ref. 22 (among others) shows rather diverse changes in respiratory function among patients with succinyl-CoA ligase deficiency. The authors should state the activities of the primary cells

they work with (both those obtained from previous publications and their newly reported sample). Similar studies in the larval zebrafish models: the somites of embryos and the subsequently mature (maturing) skeletal muscles are quite abundant and complexes I through IV activity should be easily measured.

We thank the reviewer for this recommendation, and we agree that quantifying respiratory chain activities would be an interesting experiment to pursue.

Patients 1–6 (P1–P6) are featured in the references indicated in Table 1 of the manuscript, and the effect of SCL deficiency on respiratory complexes is quite well established in these previous publications. OXPHOS deficiencies have been examined in fibroblasts in the patients with the Faroese founder mutation c.534+1G>A (reference 59). We added the published data on OXPHOS deficiency in the included patients (P1–P6) and OXPHOS analysis in the newly reported patient 7 (P7) to Table 1.

We did not measure mitochondrial respiratory complex activities in *sucla2* mutant zebrafish larvae. These experiments, although feasible, would represent a considerable investment due to the tiny size of zebrafish larvae (3–4 mm in length). Indeed, a pool of ~50 larvae and about four replicates are necessary to obtain interpretable data. This procedure would involve “head”-clipping of ~1000 larvae for genotyping, followed by pooling of the stored tails for the quantification of complex activities. Relative to the effort and the sacrifice of approximately 1000 larvae, we think that the knowledge gained would be limited, although clearly of interest.

However, we have quantified the oxygen consumption rate in *sucla2* mutant zebrafish and wild-type controls (using seahorse analyzer instruments). Larvae with *sucla2* mutation indeed have a reduced baseline and maximal oxygen consumption. Furthermore, overexpression of *sirt5* partially rescues the deficiency, which we find remarkable considering the context of loss of function of a key TCA cycle subunit. **These data are included in a new Figure 5** titled “*sirt5* restores maximal oxygen consumption rate and improves survival of *sucla2*^{-/-} zebrafish”.

2. Interface of ketone utilization and succinylation.

Although not rising to the degree of statistical significance, the rate-limiting enzyme of ketolysis (OXCT1 (3-oxoacid CoA transferase 1; formerly called succinyl CoA:3-oxoacid CoA transferase, SCOT) is succinylated and differentially expressed in their patient samples. Acetoacetate is also increased (albeit not significantly, both d0 and d5) in fibroblasts. OXCT1 catalyzes the CoA-thioesterification of the ketone body acetoacetate, taking the CoA from succinyl-CoA (as the former name SCOT indicates; and not free CoA-SH) to activate this fuel for oxidation. Since cell culture in complete medium may not reveal changes in ketone utilization, the authors might find a critical pathogenetic mechanism-defective ketone utilization- by incubating cells in low-glucose medium and adding 3-hydroxybutyrate (easier) or acetoacetate (more challenging since

sodium salts invariably decarboxylate spontaneously, and it is tricky to work with esters and anhydrides for fresh, slightly alkaline preparation and treatment of cells in culture).

The observations that OXCT1/SCOT is succinylated and that the change is mirrored by a mild, but non-significant increase in acetoacetate are interesting. The suggestion to incubate the cells in low-glucose medium that is enriched in ketone bodies is very valuable and may indeed reveal a deficiency in utilizing ketone bodies as energy fuels. Ketosis is not a typical feature of mitochondrial disease and is therefore not measured in our patients. While of clear interest, due to the lab shutdown, this hypothesis cannot be tested at this point.

3. Zebrafish tools:

- a. the molecular lesion carried by the *sucla2* mutant should be presented formally. Stating what exon was targeted and even listing the guide RNA does not provide sufficient detail (and will be difficult to inventory in zfin.org). The full deletion (as a DNA sequence and a cartoon of the chromosome and the protein if a truncation is made), and description of the heterozygous carriers should be presented.
- b. There is mention of a *sirt5*^{-/-} null mutant in the methods, but no experiments are presented with this model.
- c. Additional details regarding the Sirt5-transgenic should be presented: is this transgenic model used heterozygously (i.e., 1 integration site achieved after outcrosses)?

We agree with the reviewer that additional background information on the zebrafish tools would benefit the manuscript. We added an explanatory scheme to Figure 4 highlighting the type of mutations in the *sucla2* and *sirt5* knockout animals, as well as the overexpression construct for the *sirt5* gain-of-function experiments. Both gene disruptions are done in early exons and lead to frameshift mutations. The alignments of cDNA of wild-type and *sucla2/sirt5* mutant sequences are now provided in Supplementary Figure 5. Of note, all lines used in this study have been registered at the central repository ZFIN.org, and line designations are indicated in the methods section of the manuscript.

In addition, we included the qPCR validation of the loss of *sucla2* and *sirt5* transcripts, indicating non-sense mediated decay of the mRNA. The western blot images in the same figure further show loss of detectable *sucla2* protein. Unfortunately, we were not able to show the loss of Sirt5 protein in *sirt5*^{-/-} animals despite having tested several commercial antibodies. Taking together the qPCR data and the increased succinylation patterns in *sirt5* knockout animals, we feel confident to work with a loss-of-function model.

Furthermore, we included a new Supplementary Figure 4 highlighting changes in succinylation patterns in skeletal muscle of adult heterozygous carriers. For these experiments, we used a muscle-specific transgenic *sirt5* overexpression line. The findings are indeed interesting as even heterozygosity of *sucla2* loss-of-function is enough to increase lysine succinylation. Whether

these changes are functionally relevant remains to be determined. Also, whether these zebrafish results are relevant for humans (i.e., in recessive carriers of *SUCLA2* mutations, such as the parents of the affected children) remains to be studied.

We obtained preliminary data showing that aged *sucla2*^{+/-} zebrafish exhibit impaired swimming performance in exercise tests. We did not include these data in the Supplementary Figure as we have not yet been able to study the epistasis with Sirt5 gain and loss of function, and therefore, we feel that these data would be rather distracting in the context of the current manuscript.

In this study, we used *sirt5* knockout zebrafish to test whether loss of sirt5 function further increases protein succinylation. Taken with the gain-of-function experiment that decreased protein succinylation, these data indicate that changes in sirt5 activity influence succinylation load in both directions (Figure 4 and Supplementary Figure 6 Information). For functional studies, we focused on the effects of sirt5 gain-of-function in this study.

Of note, we have thoroughly characterized the *sucla2* mutant zebrafish. These data will be submitted to a specialized peer-reviewed journal in the near future to provide these tools to the community studying mitochondrial disease.

4. Zebrafish survival analysis.

a. Figure 4c should be presented in the standard format used in 4d (i.e., a step function, not curves with error bars-adding up cohorts of zebrafish undergoing survival analysis is acceptable, just as is rolling recruitment of human subjects for trials-not all subjects are recruited on the same day, nor are they the exact same age).

Thank you for this recommendation. We changed the layout of the graph accordingly.

b. The log-rank test approach used is not standard. Even with adjustments with the post-hoc test selected (Benjamini-Hochberg), there is substantial risk of seeing differences that arise from chance. A simpler approach is to perform the conventional log-rank tests and then perform a Bonferroni correction. The key aspect is to be transparent about how many comparisons are being made: the K needs to be on the higher end, as explained clearly in this work flow for the very popular statistical software used by the authors - this product, incidentally, should be called 'Prism' in the first column of the Antibodies and reagents table; and the source is GraphPad (what they call "Prism GraphPad is analogous to calling a unique car model and make when the column calls for just a model).

We thank the reviewer for pointing out this oversight. We re-analyzed the experiments using the statistical method recommended by the reviewer. As a result, indeed, the second experiment in volume restriction showed an alpha >0.5 but with a trend towards significance

($p=0.0146$). Due to the COVID situation, we are currently not able to repeat this experiment and decided to move this result to the supplementary information as supporting material.

c. Important controls for 4D are needed to firmly cement that acidification of the extracorporeal environment is the driver of death: in one experiment, adding 5 mM HEPES pH 7.2 (as in the classic volume restriction-live imaging study: Westerfield M , Liu DW , Kimmel CB , Walker C. (1990). Pathfinding and synapse formation in a zebrafish mutant lacking functional acetylcholine receptors . *Neuron* 4 , 867 – 874) will serve as a buffer against the acid of respiration and the acid from the excreted toxic metabolites (not just methylglutaconic acid, as ref. 22 describes-indeed, it is methylglutaconic acidemia that appears most specific for succinyl-CoA ligase deficiency in that large cohort study). A second control would involve changing the E3 medium over the course of the survival study, since this would remove toxic metabolites and acid that accumulate.

Acid accumulation in models of mitochondrial dysfunction increase acidification of volume restricted media in zebrafish (Van der Velden et al, 2011 PNAS). However, we agree with the reviewer that control experiments using a buffer would help to solidify the hypothesis that acid accumulation is indeed the driving factor of increased lethality. We are currently not able to perform lab experimentation and due to the non-significant result when re-analyzing using the Bonferroni post-hoc test, we elected to remove this experiment from the main figure. We still believe that the results are of interest for the reader as supporting material. However, we now added results that show an improved oxygen consumption rate in zebrafish larvae that overexpress *sirt5* in the background of *sucla2* deficiency. This new observation further supports the concept that a shift occurs from anaerobic to oxidative metabolism in this model. **These data are presented in a new Figure 5 together with the improved survival in standard conditions.**

Reviewer #2 (Remarks to the Author):

This manuscript by Gut et al. focuses on the interplay between lysine succinylation (suK), the mitochondrial disease succinyl-CoA ligase (SUCL) deficiency, and the lysine deacylase SIRT5. The authors find that accumulation of succinyl-CoA in context of SUCL deficiency leads to hypersuccinylation of many mitochondrial and non-mitochondrial proteins; these suK sites largely overlap with those occurring in SIRT5 KO cells. The authors generate a fish model of SUCL deficiency and show that global hypersuccinylation, and early mortality, are partially rescued in this strain by concomitant SIRT5 overexpression.

Overall, this manuscript is well-conceived and -executed, and makes a significant novel contribution to the study of lysine post-translational modifications. Moreover, it may have significant downstream therapeutic implications, given that SUCL deficiency, although rare, is a

severe mitochondrial disease with no effective therapy available currently. I recommend publication in Nature Communications, provided that the authors can address the following relatively minor concerns:

1. Given that the authors make conclusions about relative global suK levels between samples based on immunoblots, these blots (Figs. 1e, 1f, 4a, and 4b) need to be quantified.

We quantified the immunoblots from human and zebrafish samples and added this information to the main text, the figures depicting the blots, and the Supplementary Figures 2 and 3.

2. According to the legend to Fig. 1, “Controls (C1-C3) are fibroblasts from age-matched patients with other mitochondrial diseases”. The authors should also compare suK levels in a couple of truly wild-type (not mutant) fibroblast lines to SUCL-deficient cells.

We agree with the reviewer that cell lines from healthy controls would be an optimal control. The patients in the study are children, and all the samples used are originally taken for diagnostic purposes. Obtaining cell lines from patients requires a skin and/or muscle biopsy, which is an invasive procedure, and in children, it usually requires general anesthesia. Therefore, we do not have access to cell lines from healthy age-matched children controls. We also consider it most reliable to study control cultures that have been similarly established in our laboratory (patients and controls with other mitochondrial disease) as our patients. This point has now been commented on in the text under the results as well as the material and methods sections.

3. I think caution is in order when comparing the magnitude of hypersuccinylation in SIRT5-deficient mouse embryo fibroblasts compared to human SUCL-deficient cells (p. 12). Such differences may stem from species differences, differences in culture conditions, differences in cell type (embryo-derived fibroblasts versus adult skin fibroblasts), etc. These caveats should be mentioned by the authors.

We reworded the main text to make the challenges in comparing both species and laboratory protocols more evident to the reader (please also see complete comments to reviewer #3). We also performed additional analyses to add a direct comparison of all lysine sites that are affected in both species (**Figure 3 and Supplementary Figure 4**). The finding that SCL deficiency causes a higher mean increase of succinylation across these sites remains an approximation, but is supported by the experiments in zebrafish larvae that directly compare succinylation levels in both genotypes in a controlled experimental setting. We believe that showing the results of our proteomic analysis, along with a transparent message of such a comparison, is important for future studies of key regulatory enzymes/lysine residues that are consistently affected by different drivers of succinylation (or other types of acylation).

4. Overexpression of SIRT5 in the transgenic fish line should be documented, preferably by western blot.

We added a qPCR quantification of the transgenic *sirt5 mRNA* relative to endogenous *sirt5* expression levels (**Figure 4d**). Unfortunately, we have not found a suitable antibody for zebrafish *sirt5*. Together with the results showing de-succinylase activity (**Figure 4f and Supplementary Figure 6**), we believe the model is adequate to study *sirt5* gain-of-function.

5. In Fig. S2, the authors have inadvertently omitted the designation of SIRT5 overexpression.

We thank the reviewer for pointing out this oversight. We corrected it in the new Supplementary Figure related to SIRT5 overexpression.

6. One interesting question that arises from these studies is why the phenotypic effects of SIRT5 deficiency are comparatively mild, whereas SCL deficiency is a severe disorder. This may warrant attention in the Discussion section.

We believe the biochemical block of SCL flux in SCL deficiency is the dominant drivers of the clinical symptoms that are further aggravated by an accumulation of lysine-succinylation PTMs. In contrast, SIRT5 deficiency alone likely leads to a slow accumulation of protein succinylation over time that only becomes symptomatic when other cellular processes of PTM removal become insufficient (e.g., autophagy, proteasome activity). In addition, our data indicate that SCL deficiency leads to a larger extent of SuK marks than SIRT5 deficiency.

We added the following paragraph to the discussion:

“Notably, although we identified succinylation as a modifier of SCL disease, the underlying genetic defect within the TCA cycle is likely the dominant pathomechanism responsible for the severity of the clinical symptoms. The concept of carbon stress acting on top of the genetic defect may explain the occurrence of clinical symptoms at a young age in patients in contrast the age-related phenotypes observed in Sirt5 knockout mice²¹.”

7. Although I do not feel that the absence of this study is a deal-breaker for this manuscript, it would have been great to generate a SIRT5HY catalytic null transgenic fish line, to test directly whether SIRT5 activity is required for the rescue observed.

We agree that testing a SIRT5HY catalytic null gene in zebrafish would indeed be a valid control experiment. However, considering the substantial amount of time it takes to generate and work with an additional transgenic line, we prioritized other lines of investigation related to this study.

Reviewer #3 (Remarks to the Author):

The manuscript by Gut et al (SUCLA2 mutations cause global protein succinylation contributing to the pathomechanism of a hereditary mitochondrial disease) provides evidence that succinyl-CoA ligase (SCL, SUCLA2) deficiency leads to excess accumulation of succinyl-CoA, and that the resultant increase in protein lysine succinylation contributes to the disease phenotype of individuals harboring mutations in SUCLA2.

The authors compare previously published datasets of protein acetylation and succinylation in mouse liver tissue, and use quantitative mass spectrometry to analyze succinyl-CoA and succinylation levels in fibroblast and myotube cell lines from SUCLA2-deficient patients and from control individuals. Bioinformatic analyses are performed to compare increased succinylation in SCL and a previous dataset of SIRT5-regulated sites. In the penultimate experiment, the authors show that SIRT5 overexpression can suppress the lethality SUCLA2 mutant zebrafish.

While the authors provide intriguing evidence that pathological succinylation caused by increased succinyl-CoA may contribute to disease in individuals with SUCLA2 mutations, the bioinformatics analyses of protein succinylation are flawed and the data is sometimes presented in a misleading manner. The analyses of SCL deficient zebrafish is well done and supports their hypothesis, but could be better supported with additional experiments. In particular, the manuscript would be greatly improved by avoiding flawed bioinformatics analyses that are over-reliant on comparing numbers of identified sites and instead quantify how SIRT5 and SCL deficiency interact at the site level in the model systems (cell lines and zebrafish) employed in this study.

We appreciate the reviewer's critical review of our manuscript and have significantly overhauled the bioinformatic analysis as suggested. We hope that the changes detailed below effectively address these issues. We believe that the manuscript has greatly improved as a result.

Major comments:

1. Why is the analysis of previously published acetylation data included in this manuscript? The authors claim that these data show that TCA enzymes are particularly susceptible to nonenzymatic acetylation. However, there is no data presented here or elsewhere to suggest enrichment of SIRT3-regulated sites on TCA proteins compared to other mitochondrial proteins. The analysis in Supplementary Figure 1A is flawed because it only compares the number of identified sites on TCA proteins compared to all proteins. Several studies show that more acetylation (and succinylation) sites are identified on abundant proteins (PMID: 30837475, and Ref#11) due a technical bias to identify acylated peptides from abundant proteins. Since TCA

proteins are highly abundant this result is completely unsurprising, are TCA proteins enriched when compared to similarly abundant mitochondrial proteins? To perform this analysis correctly you should also correct for the lysine content of the analyzed proteins, large proteins with numerous lysines will harbor more sites than smaller proteins with fewer lysines. See also (PMID: 26513550) for gene ontology analysis that is corrected for protein abundance bias.

How does including analysis of acetylation sites contribute to this manuscript or our understanding of the role of succinylation in SCL pathology.

The analysis of succinylation in Supplementary Figure 1A is similarly flawed and should be corrected or removed.

I suggest that the authors use abundance corrected intensity (acetylated peptide intensity corrected for protein abundance by dividing by total protein intensity) to determine relative acetylation stoichiometry between different classes of proteins (for example TCA proteins versus other mitochondrial proteins, or SCL proteins versus other mitochondrial proteins). This measure is independent of protein abundance bias and can suggest higher levels of acetylation within certain classes of proteins, we first described this in yeast (PMID: 24489116) and recently used it to compare different classes of CBP/p300-regulated acetylation sites (PMID: 29804834).

Regarding the acetylation data, we agree with the comment that this dataset is not relevant to the message of this manuscript. Focusing entirely on succinylation and its relationship with Sirt5 helps to increase clarity. We removed the panels in Figure 1 and Supplementary Figure 1 and removed the mention of them throughout the text.

Regarding the bias toward detection of sites from abundant proteins, as suggested, we repeated the analysis of the prevalence of succinylation sites by dividing the # of observed succinylation sites by: protein length, # of lysines in that protein, or label-free quantification for unmodified peptides in that protein. The panel a of Supplementary Figure 1 was reworked to show those results in a histogram format. We found that when divided by the protein's abundance (based on label-free quantification (LFQ) from unmodified peptides from the corresponding protein measured separately), the TCA cycle proteins show more sites on average than the rest of the distribution. This difference in groups was significant according to a Wilcoxon sum rank test (p-value $6E-4$). This is now illustrated in Supplementary Figure 1.

The first paragraph of the results now reads:

“We and others reported that TCA cycle enzymes carry succinylation modifications in livers from Sirt5^{-/-} mice^{16,21,25}. We applied a comparative analysis approach to these datasets. Among all reported succinylated proteins, TCA cycle enzymes are enriched for succinylation modifications when all cellular proteins are rank-sorted by the number of acylated sites

corrected by the abundance of those proteins (Wilcoxon test p-value 6E-4, Supplementary Fig. 1a). Most TCA cycle subunits carry several succinylation sites (Supplementary Fig. 1b). This finding is consistent with the concept that the abundance of succinyl-CoA in the mitochondria drives non-enzymatic acylation, and that the TCA cycle is the predominant source of succinyl-CoA^{4,11}. In the absence of the desuccinylase, SIRT5 increased the abundance of many succinylation sites on TCA cycle proteins (Fig. 1b). These data suggest that TCA cycle proteins involved in succinyl-CoA production or utilization are uniquely susceptible to reversible lysine succinylation.”

2. Page 8, 164-166. The authors state “Notably, of all enzymes of the TCA cycle, SCL subunits had by far the largest fold change among Sirt5-regulated acylation sites: three subunits, Suclg1, Sucla2 and Suclg2, had several Sirt5-regulated lysine residues each (Fig. 1b and Supplementary Fig. 1c,d).”

There are several problems here. The authors refer earlier in the manuscript to 3 studies from which these data are derived, 2 of which were studies of succinylation, Rardin et al, and Park et al. First, it is unclear which of these two studies the data shown in Supplementary Figure 1c,d comes from. Most importantly, the claim that these sites have by far the largest fold change is false. By comparing the data shown in Suppl. Fig. 1d to the data from each of these studies it is clear these data come from the Rardin et al study. However, the largest change shown in Suppl. Fig. 1d is 61.6 fold for Suclg1 K94, in the data from Rardin et al, this is the site with the 12th highest increase, Atp5o is top with a 395-fold increase. The second highest site shown in Suppl. Fig. 1d is Suclg1 K66 with 25.6, which is the 32nd most increased site in their data. The data clearly does not support the claim that SCL enzymes have “by far the largest fold changes” and I assume that the problem here is a mistake in describing these data, because as currently described, it is not true, at all. Can the authors support this claim by using an actual statistical test, (for example, comparing the maximum, median, and/or average fold changes between regulated proteins), and using data from both studies?

We apologize that the source of the data was not clear, and that the wording was unclear. Indeed, this analysis is from the Rardin et al. (2013) manuscript on lysine succinylation, and we meant that these changes were the largest among TCA cycle proteins, not among all proteins. We updated the opening sentence to this paragraph to make these two points clearer.

“Based on data published by Rardin et al.¹⁷, among all Sirt5-regulated succinylation sites on the proteins in the TCA cycle, succinylation sites on SCL subunits (SUCLG1, SUCLA2 and SUCLG2) had the largest fold-change (Fig. 1a and Supplementary Fig. 1c,d).”

To support this hypothesis, we performed ANOVA to assess if the distributions of log₂(fold change) of each TCA cycle protein were different and found that, indeed, the group means

were different (p-value 1E-7). Tukey post-hoc testing revealed that the only significantly different group means were the three succinyl-CoA ligase subunits (adjusted p-value < 0.02) compared to the sites on the other TCA cycle proteins, Fh, Idh2, Dld, Sdha. We added the following text to the paper:

“Based on data published by Rardin et al.¹⁷, among all Sirt5-regulated succinylation sites on the proteins in the TCA cycle, succinylation sites on SCL subunits (SUCLG1, SUCLA2 and SUCLG2) had the largest fold-change (Fig. 1a and Supplementary Fig. 1c,d). Distributions of succinylation sites on TCA cycle proteins were different from each other (log2(fold change), ANOVA p-value = 1E-7); succinylation site changes on all three SCL subunits (SUCLA2, SUCLG1, SUCLG2) were statistically significantly higher than sites on the other TCA proteins FH, IHD2, DLD, and SDHA (Tukey’s post hoc test adjusted p-value < 0.02).”

3. The authors claim that sites on *suclg1* and *sucla2* occur within the structure of these proteins at positions that are likely to impact their enzymatic function. However, the stoichiometry (degree of modification) at nonenzymatically acetylated lysine residues has been shown to be extremely low. Since the manuscript hypothesizes that nonenzymatic succinylation drives (some of) the disease phenotypes, can the authors please determine the stoichiometry of modification using peptide standards, either in SIRT5 and/or SCL deficient cells or tissues. These measurements would be important to support the claim that “These data suggest that the remaining SCL activity in mutations leading to hypomorphic phenotypes can be further reduced by succinyl-CoA-mediated auto-succinylation.”

We appreciate the suggestion to measure the stoichiometry, and normally, we would comply with this request. Given the current lab situation, we cannot perform additional experiments. Since the claim that additional modification of SCL could be important for further SCL inactivation by succinylation is not central to our manuscript, we hope that instead greatly softening this claim will suffice:

“These data suggest that succinylation sites on SCL are important targets of regulation by Sirt5, and that succinylation sites on SCL could function to inhibit SCL as part of a feedback loop.”

4. Fig 2g and Page 10 215-219

It is incorrect to compare a group of modified proteins to all proteins in gene ontology analysis as shown by (PMID: 26513550). This analysis only shows the abundance bias of succinylation detection. Furthermore, since the process is nonenzymatic we expect all proteins to be succinylated, which is consistent with the observation that all (98%) of sites were increased in patient cells. Why then perform gene ontology analysis if all proteins are affected? (but you only detect some of them because they are more abundant). It would be more informative to compare the group of proteins showing the greatest increases in succinylation to those showing

the least increase in succinylation (If possible since this likely varies at the site-level and not the protein-level).

We agree that we cannot conclude the true groups of proteins that are modified from this analysis, but we think it is useful to summarize the proteins on which we detected significant succinylation fold-changes. As suggested, we compared the proteins that were most affected to those that were least affected. As predicted by the reviewer, this was not possible, because proteins have both high and low fold-change sites. We changed the text to report the observation that these proteins are likely the most abundant proteins more explicit, and to focus on that we detected these rather than that these are the true population of modified:

“We performed gene ontology (GO) term enrichment analyses using the list of succinylated proteins found in fibroblasts to identify the subset of proteins that harbor increases in lysine succinylation due to SCL deficiency. Although this analysis does not reveal the true population of modified proteins because of analytical bias toward detecting sites on the most abundant proteins (here, we cite Scholz, C. et al³⁰), this analysis does reveal the types of proteins we followed in our study. As expected, the TCA cycle GO term was significantly enriched at the top of the list (Fig. 2g). Our data also overrepresented succinylation sites on proteins involved in mitochondrial energy production, such as ATP biosynthesis, pyruvate metabolism and beta-oxidation. Additionally, GO terms representing proteins outside the mitochondria were enriched, such as cell-cell adhesion, protein folding, and glycolysis/gluconeogenesis.”

5. Fig 2h

The figure sorts proteins based on the number of detected succinylation sites. As stated above in major comment #1, this analysis is compounded by the technical bias to identify a greater number of sites on abundant proteins and is further compounded by the number of lysine residues on those proteins and whether they can be detected as tryptic peptides in the MS. Furthermore, we expect that all solvent accessible lysines are nonenzymatically succinylated, so there is no point in this analysis. The analysis is flawed.

We thank the reviewer for the comment and agree with the important point that our detection is biased by the abundance of the proteins in the sample. However, we assert that the previous interpretation was flawed, not its analysis. We believe it is interesting for readers to know what sites we could detect even if they do not represent the true population of sites that are influenced (which are likely all the sites). We adjusted the text to make this important caveat clear:

“A plot of the top 20 proteins sorted by number of lysine marks (Fig. 2h) and fold-change of individual lysine residues (Fig. 2i) showed that we detect many succinylation sites on proteins that reside in different subcellular compartments, including mitochondria, the cytosol and the endoplasmic reticulum. It is important to note that this subset of sites almost certainly is biased toward the more abundant proteins in the sample and therefore does not reflect the true population of sites influence by SCL deficiency. In fact, since nearly all succinylation sites

increased and we expect this process to be nonenzymatic, there should be no enrichment in functional categories. Still, these analyses of which proteins we detected are presented to convey our analytical coverage.”

p11, 223-225

“In summary, succinylation in response to high levels of reactive succinyl-CoA affects proteins of diverse cellular pathways, and the TCA cycle and metabolic pathways are major targets.”
The TCA and metabolic pathways are not necessarily major targets, it is just easier to identify succinylation sites on these abundant proteins.

We removed the statement that the TCA cycle and metabolic pathways are the major targets, and instead conclude this paragraph with a more general statement:

“In summary, SCL deficiency leads to high levels of reactive succinyl-CoA that affect proteins of diverse cellular pathways by non-enzymatic lysine succinylation.”

6. Fig3A

The presented overlap in observed proteins between SUCLA2 deficiency and SIRT5 deficiency is misleading. While many of the same proteins were identified in these two studies, only 124 sites on 48 proteins were found in both studies. This explains why the authors fudge the following analyses (Fig3B-E) by comparing the numbers of succinylation sites found on proteins instead of comparing the actual changes occurring at the same sites in both experiments. The site-level overlap for these studies is only 13% for SCL sites, compared to the 51% they show for proteins in the figure, this is misleading.

The overlap between proteins in the dataset was re-assessed more carefully. Uniprot accessions from Park et al. Table S1Bii were mapped to gene names and mouse protein sequences using the R package Uniprot.Ws (see supplemental code 1). Since the lengths of mouse and human proteins differ due to evolution, we performed pairwise sequence alignment between the common succinylated mouse and human protein sequences and adjusted the mouse sequence residue numbering, based on the observed insertions and deletions. This revealed slightly better overlap of 238 sites on 102 proteins. To better reflect this overlap, we updated Figure 3A to include two Venn Diagrams, one with protein overlap and one showing site overlap.

We added a second Venn Diagram showing the site-level overlap between the two studies as a new panel Figure 3B clarifying that the original Venn Diagram describes the protein level. We also updated the text in this part to remove extrapolations in interpreting the analytical overlap:

“To better assess potential overlap between these two drivers of protein succinylation, hyper-succinylated proteins and sites detected in our fibroblasts from patients with SCL deficiency

were compared with those found in SIRT5 KO mouse embryonic fibroblasts by Park et al. 21. At the protein level, 102 proteins carry at least one succinylated lysine in both conditions, representing 41% of all proteins detected with hypersuccinylation due to SCL deficiency, or 29% of hyper-succinylated proteins detected due to deletion of SIRT5 (Fig. 3a). Since the two compared studies were carried out in different organisms, and mouse protein sequences carry amino acid insertions and deletions relative to human proteins, pairwise sequence alignment of these 102 common proteins was performed to allow site-level comparison. This revealed that 238 sites were in common between the two studies (Fig. 3b). Interestingly, the extent of hyper-succinylation due to SCL deficiency was greater than was observed from mouse embryonic fibroblasts lacking SIRT5 (Fig. 3c)."

7. Fig3C and D

As stated above several times, several studies have shown that we identify more sites per protein on abundant proteins (the authors are welcome to determine protein abundance in their own samples, which is easy, and verify this bias in their own data). That the authors identified a large number of sites on the same proteins as in a study of SIRT5 deficiency is both unsurprising and uninformative. What is the significance of this finding? The same relationship would almost certainly occur if you compared normal fibroblasts from mice to fibroblast from human. This type of control analysis is lacking, and a technical bias is overinterpreted. Sadly, there is plenty in this manuscript that is interesting without relying on technically flawed analyses such as this.

First, please note that we did determine protein abundances in our samples, and we used those abundance measures to correct any site-level change (see methods section Proteomic data analysis):

"Relative quantities of proteins were used to correct changes in observed PTM abundance before statistical comparison of quantities between patient samples and controls with mapDIA 45."

Although we agree that there is analytical bias toward the most abundant proteins, this analysis gives a sense of the analytical overlap, which cannot be taken for granted when comparing any two studies. Therefore, we believe that the referenced figure panels and underlying analyses are informative in an analytical context. We reworded the description of these results to remove any extrapolation to the true population of succinylated sites and better convey a focus on the analysis overlap:

"Although the sites and proteins we detect in our analysis are biased toward the most abundant succinylated sites and proteins, another useful way to compare the analytical overlap between studies of SCL deficiency and SIRT5 KO is by comparing the number of modifications detected per protein. Among these common proteins, the number of succinylation sites per protein was

correlated between both conditions ($r^2=0.5124$), and FLNA, MYH9, and HADHA had the most succinylated sites (Fig. 3d)”

8. Fig3B and E

It is an interesting finding that SCL patients show more substantially increased succinylation at a majority of sites compared to SIRT5 deficiency (although SIRT5 deficiency causes more dramatic increases at a subset of sites). However, the authors miss a chance to investigate how SIRT5-regulated sites are impacted in SCL. Are SIRT5-regulated sites increased in SCL to the same degree as all sites? In other words, does SIRT5 suppress increased succinylation in SCL? This is particularly relevant question given the results shown in the Zebrafish model. However, I assume that the authors are unable to address the question since only 124 sites are quantified in both studies, thus the authors are comparing different sites when performing this analysis. It would be best if the authors KO'd SIRT5 in this SCL cell lines to directly investigate the interaction between SIRT5 and SCL deficiency.

In Fig3E, please indicate sites that were common between the studies by connecting them with lines, this would indicate the trend between SIRT5 regulation and SCL impact at individual sites.

We agree that the comparison of specific fold-changes at each site is more interesting, and we **included two additional analyses in the supplemental information as well as one additional supplemental table** to enable these comparisons for readers and reviewers. As mentioned above, we repeated this comparison of sites with aligned sequences between the species and found better overlap of 238 total sites. We reworded the results text to read:

“Although the comparison of the two different species and methodologies does not allow completely quantitative interpretation, the median fold-changes across the differential succinylation of most of the top 11 proteins were higher in SCL deficiency than in SIRT5 deficiency (Fig. 3f). Comparing the exact sites of these same 11 proteins between these two conditions makes clear that almost all sites had lower fold-changes in the SIRT5 deficiency (Supplementary Figure 4). Notably, some proteins had sites with more similar regulation between these two studies, such as citrate synthase and HADHA (Supplementary Figure 4, Supplementary Table 6).”

9. P12, 256-260

“These data show an extensive overlap between proteins that accumulate lysine succinylation marks in response to SUCLA2 and SIRT5 deficiencies and suggest non-random succinylation events independent of whether succinylation is derived from increased succinyl-CoA levels or

decreased lysine de-succinylation activity.”

This statement is not supported by the data and is misleading.

- A. All proteins (98%) show increased succinylation in SCL, which is expected based on the nonenzymatic mechanism, so there is no special significance that these sites overlap with SIRT5 regulated sites (if all sites are regulated, they will overlap with any sites found in any study).
- B. The only thing that is non-random is that the authors employed antibody enrichment of succinylated peptides that is similarly biased for peptides from abundant proteins. To show that this is indeed non-random the authors would need to correct for protein abundance biases, or perform a similar comparison of patient derived normal human fibroblasts to normal mouse fibroblasts.
- C. Since the analysis relies on comparing the number of sites found on each protein it says nothing about the relationship between degree of SIRT5 regulation and degree of succinylation in SCL. This would be much more informative.

We agree with Brian’s points and changed the text to reflect these points:

*“Because nearly all (98%) of succinylation sites increased due to SCL deficiency, as would be expected in a non-enzymatic process, the sites we detect in our study are likely to overlap with those found in other datasets related to hyper-succinylation. Therefore, examination of the exact fold-change resulting from each process on individual lysine sites found in both datasets is of interest. Although the comparison of the two different species and methodologies only allows an approximation of fold-changes, we estimate that the median fold-changes across the differential succinylation of most of the top 11 proteins were higher in SCL deficiency than in SIRT5 deficiency (**Fig. 3e**). Comparing the exact sites of these same 11 proteins between these two conditions suggests that almost all sites had lower fold-changes in SIRT5 deficiency compared to SCL deficiency (**Supplementary Fig. 4**). Notably, some proteins had sites with a similar regulation between these two studies, such as citrate synthase and HADHA (**Supplementary Fig. 4, Supplementary Table 6**). These data show an extensive analytical overlap between proteins detected to accumulate lysine succinylation marks in response to SUCLA2 and SIRT5 deficiencies, which cause increased protein succinylation through increased succinyl-CoA levels or decreased lysine de-succinylation activity, respectively.”*

10. Fig4C and D

Why were *sucla*^{-/-}, *sirt5*^{-/-} animals not analyzed? If SIRT5 protects against increased succinylation we would predict that these double mutant animals would show decreased

survival compared to *sucla*^{-/-} animals. The data for these experiments should also be shown and the results discussed.

We used *sirt5*^{-/-} animals to analyze the effects on protein succinylation in wild-type and *sucla2*^{-/-} and *sucla2*^{+/-} genetic backgrounds. For functional studies, we focused on *sirt5* gain-of-function to address the question of whether reduction of the succinylation load has benefits in the disease model. We agree that the reverse experiment testing of whether *sirt5* loss-of-function further aggravates the phenotype is of interest. We have done initial experiments in this direction. The results were inconclusive as it is difficult to quantify changes in parameters related to deficiencies (e.g., OCR, survival) that are already quite severe in the *sucla2* knockout model. However, we agree that NAD⁺ decline in SCL patients could lead to secondary *Sirt5* deficiency and affect lysine succinylation events and their functional consequences. This aspect is mentioned in the discussion.

Minor comments:

1. Page 4.

“This prediction was further validated experimentally in vitro 4-7.”

The authors should at least cite (PMID: 23946487) and (11) which were the first studies to show this in vitro and were published at about the same time.

We apologize for this oversight. We added both references to the sentence.

2. Page 5, 106-107

“Research with yeast indicates that disrupting the succinyl-CoA ligase reaction increases global protein succinylation 21.”

The citation is incorrect, the cited paper did not investigate yeast. The citation you want is (11).

We corrected this mistake.

3. p11, 237-240

“The similar outcomes of these different analyses suggest specificity for a subset of cellular proteins independently of the succinylation driver (i.e., both the loss of SIRT5 and elevated succinyl-CoA levels, as shown in the present study) converge on a similar set of target proteins.”

A major premise of this paper is that succinylation is nonenzymatic. What is the basis of the proposed specificity? Proximity? Unfortunately, none of the included bioinformatic analyses showed this. Also, since all sites are impacted by SCL, they will converge on any group of succinylation sites that you compare these data to.

We agree that the overlap is partially driven by analytical limitations of what proteins are detectable based on their abundance, and we agree that we cannot know from our data and analysis the basis of the modification specificity. We deleted this statement and added a statement later in the manuscript clarifying that these sites we detect will overlap with those identified by other groups using mass spectrometry, such as cited in Park et al. (2013).

4. p13, 280-282

“Combined deficiency of sirt5 and sucla2 further increased global protein succinylation, suggesting that succinylation is driven in vivo both by succinyl-CoA and by sirt5 (Fig. 4a).”

Can you provide quantification? The Western blot images are not particularly convincing.

These blots were quantified as requested and that analysis now appears in the new Figure 4e and 4f. This analysis reveals that the differences are significant.

5. p16, 344-355

“The majority of succinylated proteins we identified are located within the mitochondria and affect bioenergetic pathways, including the TCA cycle, ATP-synthesis, and beta-oxidation. Less expected was the observation of hyper-succinylation on proteins in pathways outside mitochondria. Proteins responsible for cell-cell adhesion, branched-chain amino acid catabolism and glycolysis were also found to be enriched for succinyl-lysine marks. The sources of these lysine-succinylation reactions are not clear, although increased levels of succinyl-lysine modifications have been described in conditions of SIRT5 loss-of-function 21. In SCL deficiency, the driver of succinylation is the built-up of succinyl-CoA due to mutations in SUCLA2 and, therefore, reduced enzymatic flux. Cytosolic succinylation in this context could be caused by a leakage of succinyl-CoA through damaged mitochondrial membranes or secondary SIRT5 deficiency due to depletion of NAD⁺ levels, which has been described in other forms of mitochondrial dysfunction or disease 12,37.”

Although succinylation is biased to mitochondrial proteins in human cells and mouse liver (11 and PMID: 26513550), succinylation sites are found outside of mitochondria in the absence of pathologies such as SCL or SIRT5 deficiency (11, 21 and PMID: 26513550), therefore the authors should avoid using pathological explanations for this observation.

We agree that succinylation occurs outside the mitochondria. However, what we wanted to point out is that the source of additional succinyl-CoA in the context of SCL deficiency is generated within the mitochondria, but nevertheless, many proteins outside the mitochondria show large fold-changes in lysine succinylation. Therefore, the term of “pathological succinylation” in this particular context. We rephrased this paragraph.

“In SCL deficiency, the driver of succinylation is the intra-mitochondrial build-up of succinyl-CoA due to mutations in SUCLA2 and, therefore, reduced enzymatic flux. Because succinyl-CoA is a negatively charged molecule and should not cross the mitochondrial membrane, the mechanism of lysine hyper-succinylation outside the mitochondria is not clear. Although pathways for transfers of acyl-groups across the mitochondrial membrane have been described for other acyl-CoA moieties, no such mechanism has been reported for succinyl-CoA^{37,38}. Cytosolic succinylation in this and other contexts could be caused by a leakage of succinyl-CoA through damaged mitochondrial membranes, an unidentified transport mechanism for activated succinyl groups such as succinyl-carnitine, and secondary SIRT5 deficiency due to depletion of NAD⁺ levels, the latter of which has been described in other forms of mitochondrial dysfunction or disease^{13,39,40}. “

Reviewed by Brian Weinert

Reviewer #4 (Remarks to the Author):

The authors reported on their study hyper-succinylation due to SUCLA2 deficiency with investigations into countering effects of sirtuins. This is an important area of research to improve our understanding how protein acylation affects key cellular energy pathways. The authors highlighted the elevated levels of protein succinylation and buildup of succinyl-CoA, while only moderate recovery from elevated level was observed in sirtuins expression. The authors did an excellent job in reporting their finding as the manuscript was easy to read and understand.

We appreciate the positive feedback from the reviewer and thank him/her for highlighting the importance of studying the relationship of protein acylation and energy metabolism.

Here are a few revisions and suggestions:

- Lines 13-27, punctuation inconsistencies within the affiliations.
- Line 458, “Formic Acid” does not need capitalization.
- Lines 493-494, remove “and detection”.

We made changes throughout the manuscript accordingly.

- Line 487, Ref 47 described six different extraction methods, need to indicate extraction solvent(s) used.

We apologize for the ambiguity. We added the solvent to the methods section, and we also added references to two methods paper that detail the experimental procedure (Liu et al, Anal Chem 2014 and Liu et al, JOVE 2014).

-Line 509, how much standard was added?

We used homemade 50 µL of ¹³C-fully labelled yeast extract per sample as internal standard for our metabolomics measurements of zebrafish (shown in the Supplemental Materials).

-Line 517, "+ focused liquid chromatography system" is not needed.

We removed this wording.

-Metabolite supplemental, include the more information concerning the annotation. The annotation based only on accurate mass and the accurate mass of the ¹³C signal from the Yeast biomass. Add the exp m/z (analyte and standard) and RT to the spreadsheet. The chirality is not usually determined, so that should be removed from the names. Some of the formulas reported don't seem likely candidates.

For zebrafish metabolism, the annotation of the metabolites was done by: accurate mass, confirmation of retention time and mass with its corresponding ¹³C- signal as an internal standard, as well as by authentic standards. All metabolites described were measured by negative ion mode (see Table below). Succinyl-CoA was measured by its doubly charged species; given its absence in the internal standard, it was normalized to the chemically related ¹³C-acetylCoA signal.

compound	m/z	RT (min)	comment	internal standard	m/z	RT (min)	comment
citrate	191.019	10.8		¹³ C citrate	197.039	10.8	
a-ketoglutarate	145.014	9.62		¹³ C a-ketoglutarate	150.03	9.62	
succinate	117.019	9.35		¹³ C succinate	121.032	9.35	
succinyl-CoA	432.558	10.6	double charged [M-2H]	¹³ C acetyl-CoA	415.094	8.35	double charged [M-2H]

For metabolomics on human fibroblasts, the workflows were as described in previous publications (Liu et al, Anal Chem 2014 and Liu et al, JOVE 2014). Acyl-CoA species were measured using the method described in Liu et al, MCP 2015. If available, authentic standards are used to confirm the retention time. m/z retention times and accurate mass are listed in these publications and also included in the supplementary data. When multiple isomers may co-elute, these isomers are combined as one feature with the annotation of these isomers or chemical formula.

-Similar to the authors sharing of the proteomic data, the metabolomics data could also be loaded into a public repository like Metabolomics Workbench.

Thank you for this suggestion. We uploaded the metabolomics dataset from human fibroblasts at D0 and D5 to www.metabolomicsworkbench.org, mwTab Identifier: xliu68_20200611_073356

REVIEWERS' COMMENTS:

Reviewer #1 (Remarks to the Author):

The authors have addressed my concerns. This revised study advances the field.

Amnon Schlegel

Reviewer #2 (Remarks to the Author):

The changes made to the manuscript by Gut and colleagues have substantially strengthened it, and I now support publication in Nature Communications. One minor point: the authors state that sirt5 overexpression significantly improves mean lifespan in sucl-deficient fish (Fig. 5E), but it appears that the designation of significance has been inadvertently omitted from the figure.

Reviewer #3 (Remarks to the Author):

The authors have addressed my concerns in a satisfactory manner.

best regards, Brian Weinert

Reviewer #4 (Remarks to the Author):

The authors improved the manuscript with the latest revisions and clarifications.